# Pixel super-resolved virtual staining of label-free tissue using diffusion models

Yijie Zhang[1,2,3,5], Luzhe Huang [1,2,3,5], Nir Pillar [1,2,3], Yuzhu Li[1,2,3], Hanlong Chen[1,2,3] & Aydogan Ozcan [1,2,3,4] ✉

Virtual staining of tissue offers a powerful tool for transforming label-free microscopy images of unstained tissue into equivalents of histochemically stained samples. This study presents a diffusion model-based pixel super-resolution virtual staining approach utilizing a Brownian bridge process to enhance both the spatial resolution and fidelity of label-free virtual tissue staining, addressing the limitations of traditional deep learning-based methods. Our approach integrates sampling techniques into a diffusion model-based image inference process to significantly reduce the variance in the generated virtually stained images, resulting in more stable and accurate outputs. Blindly applied to lower-resolution auto-fluorescence images of label-free human lung tissue samples, the diffusion-based pixel super-resolution virtual staining model consistently outperforms conventional approaches in resolution, structural similarity and perceptual accuracy, successfully achieving a pixel super-resolution factor of 4-5×, increasing the output space-bandwidth product by 16-25-fold compared to the input label-free microscopy images. Diffusion-based pixel super-resolved virtual tissue staining not only improves resolution and image quality but also enhances the reliability of virtual staining without traditional chemical staining, offering significant potential for clinical diagnostics.

Generative Artificial Intelligence (AI) models have achieved considerable advances over the last decade and wide applications in various fields. These models fostered the emergence of computational pathology[1], showcasing unprecedented performance in image transformation[2-14], segmentation[15,16], and reconstruction[17-19]. As one of the state-of-the-art generative techniques, diffusion models have shown a strong ability to approximate multi-modal distributions[20-22] with the versatility to be conditioned through multiple forms of guidance, including text and image[22-24]. There have been synergetic studies on applying diffusion models in computational pathology. For example, diffusion models have been demonstrated to generate photorealistic histopathology images, given guidance of texts from pathology reports[25], segmentation masks[26], RNA-sequencing data[27], domain knowledge and tissue

genomics[28]. Researchers have also studied image translation between multiple histopathological image domains[29,30], a process termed stain transformation. This includes, for example, transforming a histopathological image stained with Hematoxylin and Eosin (H&E) into an immunohistochemistry (IHC)-stained image of the same tissue slice. In addition, diffusion models have been explored for histological image enhancement and segmentation to facilitate downstream analysis and diagnosis[31-34].

To better adapt to conditional image generation tasks, researchers have also exploited stochastic bridges connecting two image domains and demonstrated various conditional diffusion models[35-37]. Among them, the Brownian bridge is one of the well-known and widely utilized stochastic processes that stems from the standard Brownian

[1]Electrical and Computer Engineering Department, University of California, Los Angeles, CA 90095, USA. [2]Bioengineering Department, University of California, Los Angeles, CA 90095, USA. [3]California NanoSystems Institute (CNSI), University of California, Los Angeles, CA 90095, USA. [4]Department of Surgery, University of California, Los Angeles, CA 90095, USA. [5]These authors contributed equally: Yijie Zhang, Luzhe Huang. ✉e-mail: ozcan@ucla.edu

(diffusion) process and is conditioned on both the start and end states. Instead of using the standard Brownian motion in the common forward diffusion process that converges to white noise, the Brownian bridge diffusion model (BBDM) learns the mapping from the target image domain to the input (conditional) image domain via a Brownian bridge[35]. BBDM has been reported to outperform standard diffusion models in various image restoration and translation applications[35,38,39]. Nevertheless, all diffusion models inherently generate outputs with relatively high variance (from run to run) compared to some of the existing generative models, including e.g., conditional Generative Adversarial Networks[40–42] (cGANs); such stochastic image variations for the same specimen raise concerns regarding their impact on biomedical image synthesis or reconstruction tasks, especially for potential uses in clinical diagnosis.

In this work, we introduce a diffusion model-based pixel super-resolution virtual staining (VS) model that transforms lower-resolution auto-fluorescence (AF) microscopy images of label-free tissue samples into pixel super-resolved brightfield images, digitally matching the histochemically stained higher-resolution images of the same tissue samples without the need for traditional chemical staining. This diffusion-based pixel super-resolved VS model significantly outperforms traditional VS methods that process the same lower-resolution AF images of label-free tissue samples, and it drastically reduces inference variance from the diffusion process, converging to stable and accurate image inference that matches the histochemically stained higher-resolution brightfield images of the same tissue samples. Our approach is built on an image-conditional diffusion model leveraging the Brownian bridge process (Fig. 1b) to effectively integrate the lower-resolution conditional image and the noise estimation from an attention-based U-Net (Fig. 1c, f, g) that incorporates the time step information to reconstruct a higher-resolution histological image—performing two tasks at the same time: (1) spatial resolution enhancement and (2) virtual staining of label-free tissue. The term "pixel super-resolution" in our context should not be confused with nanoscopy techniques that beat the diffraction limit of light. In this work, pixel super-resolution or super-resolved images refer to the capability of the VS model in synthesizing brightfield equivalent stained images with higher spatial resolution compared to the input label-free images, hence increasing the space-bandwidth product and the effective number of useful pixels in the VS images—all within the diffraction limit of light. In fact, the usage of the term super-resolution[43–45] predates the invention of nanoscopy techniques and is widely recognized within the computer vision and computational imaging communities as a benchmark task[46–49].

In comparison with other VS models, our conditional diffusion model generates better VS images with higher resolution and image fidelity matching the ground truth histochemically stained brightfield images. Besides, to mitigate the inherent high variance of diffusion models for VS applications in pathology, we introduce sampling process engineering techniques, i.e., the mean and skip sampling strategies as illustrated in Fig. 1d, e, h, i. Based on the analysis of the posterior sampling variance over time steps ($t$) as shown in Fig. 1j, we select exit points and remove the additive random noise in the following sampling steps or skip to the estimated value at $t = 0$, which significantly enhances the VS fidelity and reduces output image variance. We also introduce a post-sampling averaging strategy, which can be combined with the aforementioned sampling process engineering techniques to further reduce the output variance and improve the utility of diffusion-based VS techniques in pathology.

Virtual tissue staining using AI is critically important because it eliminates the need for chemical reagents, reduces tissue processing time and costs, and enables non-destructive, high-resolution analysis of tissue samples, paving the way for faster, more scalable diagnostics and unlocking new possibilities for digital pathology and precision medicine. This work not only presents a powerful generative model for

pixel super-resolution virtual tissue staining tasks that surpasses traditional deep learning-based VS models, but also introduces sampling process engineering techniques that provide enhanced control over diffusion model image outputs during the testing without the need for retraining or fine-tuning of the model, offering significant benefits in biomedical imaging and related applications including digital pathology.

## Results

### Pixel super-resolved virtual staining of unlabeled tissue sections using a diffusion model

The workflow of diffusion model-based virtual staining is illustrated in Fig. 1a. We began by capturing label-free AF images of unstained human lung tissue samples. Then, these slides were sent for histopathological H&E staining (for generation of ground truth stained samples), followed by digital imaging with a brightfield optical microscope to obtain the corresponding histochemical images, which serve as our ground truth images for the training and testing phases. Each one of the label-free AF images was paired and registered with respect to the corresponding labeled histochemical images to create the training/testing dataset for virtual staining. The data capturing and preprocessing steps are detailed in the "Methods" section.

Diffusion model training and sampling represent two directions of propagation of the same stochastic process. In our approach, the forward process is modeled by a Brownian bridge, as illustrated in Fig. 1b, which is a Gaussian process with a linearly scheduled mean from $x_0$ to $x_T$ and a quadratically scheduled variance with respect to the time step[35,50]. Here, the set of $x_0$ represents the target image domain, corresponding to histochemically stained images, while the set of $x_T$ denotes the input image domain, comprising lower resolution autofluorescence images that went through a shallow convolutional neural network (used for dimension matching), as illustrated in Fig. 1a. In contrast, the reverse process is a step-wise denoising process agnostic of the ground truth image $x_0$. However, the exact distribution of $x_t$ conditioned on $x_T$, $\forall t < T$ is intractable and therefore a neural network is employed to estimate the posterior mean of $x_t$ conditioned on $x_T$ (see the detailed derivations in the "Methods" section and Supplementary Note 1). As showcased in Fig. 1f, a U-Net-based denoising network is trained to estimate the difference between $x_t$ and $x_0$, where $t$ is an arbitrary, random time step between 0 and $T$. After the training, the denoising network predicts the difference between the current state $x_t$ and $x_0$, as showcased in Fig. 1g. Notice that an additional posterior noise with variance $\widetilde{\delta}_t$ is introduced to match the variance schedule of the Brownian bridge. The posterior variance $\widetilde{\delta}_t$ is illustrated in Fig. 1j, where a large posterior noise is added to diversify the distribution; this choice, however, also introduces excessive randomness in the inference stage, which presents a challenge to achieving high consistency among multiple inferences—an important feature to have for VS applications since we need to virtually stain the tissue sample of "the patient". To mitigate this challenge of diffusion-based VS models, we engineered sampling strategies and introduced the mean and skip diffusion sampling methods, illustrated in Fig. 1h, i. The mean sampling strategy eliminates the posterior noise addition in the final few sampling steps from an empirically defined *exit* point $t_e$, and the skip sampling strategy estimates $x_0 | x_t$ directly. Detailed sampling algorithms for these two inference strategies are elucidated in the "Methods" section.

We validated the pixel super-resolution virtual staining performance of our diffusion-based VS model on human lung tissue histomorphology and compared its performance to that of a cGAN-based state-of-the-art VS method[1–3,5,6,9,13,14,51]. For a direct comparison between the two approaches, we separately trained multiple pixel super-resolution VS models (covering different pixel super-resolution factors) using our diffusion-based VS approach as well as the traditional cGAN approach (see the training details in the "Methods" section). We

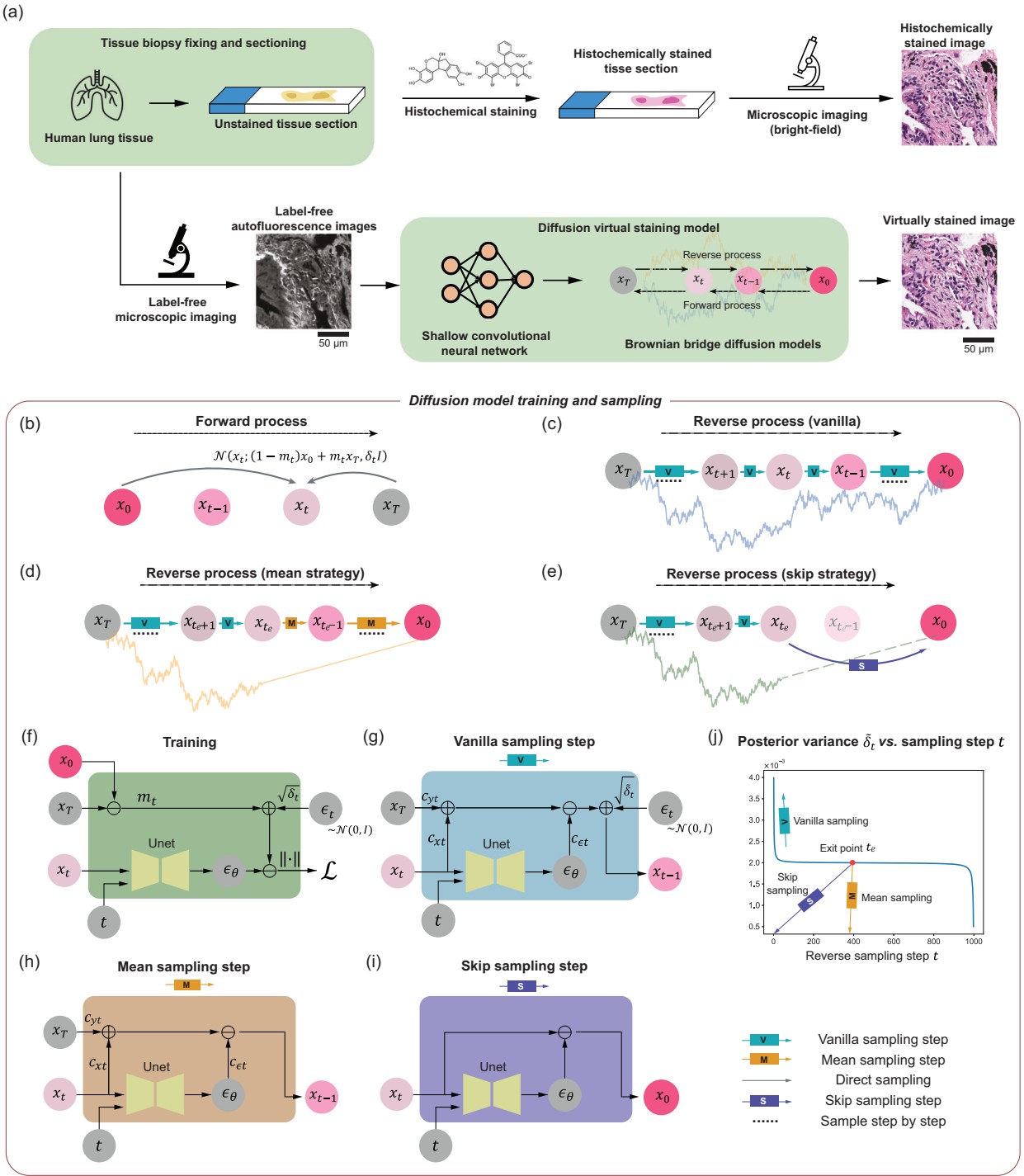

**Fig. 1 | Diffusion model-based super-resolution virtual staining of unlabeled tissue sections. a** The diffusion model-based virtual tissue staining pipeline. Our image-conditional VS diffusion model is designed based on the Brownian bridge process for both the forward and reverse processes. **b** Schematic diagram of the forward process of our Brownian bridge diffusion model. **c–e** Three different reverse sampling processes: vanilla, mean, and skip sampling strategies. **f** Detailed workflow for training the diffusion-based VS model. **g–i** Workflow for a single vanilla, mean, and skip sampling step, used individually or in combination in the reverse sampling process in (**c–e**). **j** Plot of the posterior variance $\tilde{\delta}_t$ added during the reverse sampling process, plotted against the reverse sampling step $t$, where $t = 0$ marks the end of the reverse sampling process.

termed the trained diffusion-based model as $D_x$ and the trained cGAN model as $G_x$, where $x$ denotes the pixel super-resolution factor. To span different levels of resolution loss, the input AF images underwent pixel binning for spatial undersampling, covering pixel super-resolution factors of 1× to 5× in each lateral direction; for example, a pixel super-resolution factor of 5× indicates a space-bandwidth reduction of 25-fold at the input AF microscopy images of label-free

tissue. During the blind testing phase, both approaches (diffusion vs. cGAN) were applied to 180 autofluorescence image sets (with each autofluorescence channel having 960 × 960 pixels from 15 unlabeled lung tissue sections that were never used during training to generate VS images at various pixel super-resolution factors, as shown in Fig. 2a. Our blind testing results revealed that, across all the pixel super-resolution factors (2× to 5×), the diffusion-based VS models using the

mean sampling strategy consistently outperformed cGAN-based models; this performance advantage can be visually confirmed in Fig. 2a and was quantitatively demonstrated in Fig. 2b by the higher structural similarity index measure (SSIM)[52] values, lower learned perceptual image patch similarity (LPIPS)[53] values, and higher peak signal-to-noise ratio (PSNR) values calculated with respect to the high-resolution brightfield images of the same histochemically stained tissue samples (our ground truth). Specifically, as indicated in the arrowed region of Fig. 2a, due to spatial resolution loss of the AF microscopy images of label-free tissue samples, cGAN-based models failed to reconstruct stained regions of anthracotic pigment, which are important in lung pathology for disease diagnosis. In contrast, our diffusion-based pixel super-resolution VS models consistently stained these black pigments, presenting a good match to the histochemically stained images. This aligns with the comparison of the diffusion model-generated images and their matched histochemical counterparts, conducted by a board-certified pathologist (N.P.), which demonstrated complete concordance across all image subsegments (fibro-collagenous stroma, blood vessels, immune cells, and anthracotic pigment).

To further quantify the advantages of diffusion-based pixel super-resolution VS models, we tested the statistical significance of potential improvements observed in the SSIM and LPIPS values of the two methods (see the "Methods" section for details). The $t$-scores illustrated in Fig. 2c highlight the statistically significant performance improvements of the diffusion-based pixel super-resolution VS models over traditional cGAN-based models for pixel super-resolution factors of 2× to 5×; for the 1× case without pixel super-resolution, both of these approaches perform similar in virtual staining image quality, without a statistically significant difference between them.

To highlight the pixel super-resolution capabilities of the diffusion-based VS model, a spatial frequency spectrum analysis was performed on the input autofluorescence DAPI images, the virtually stained images generated by the diffusion model, and their corresponding histochemically stained ground truth images for all pixel super-resolution factors. The results are presented in Supplementary Fig. 1a; also refer to the "Methods" section for details. The cross-sections of the radially averaged power spectra[54,55], as shown in Supplementary Fig. 1b, reveal a significant enhancement in the spatial frequency spectra of the VS images compared to the lower-resolution autofluorescence inputs. Notably, the spectra of the VS images have a good alignment with those of the histochemically stained ground truth, underscoring the diffusion-based VS model's ability to enhance spatial resolution.

**Diffusion sampling engineering techniques for pixel super-resolved virtual staining of label-free tissue**
We compared the performance of three different sampling strategies—named vanilla, mean, and skip methods (see Fig. 1c–e)—using the VS diffusion model trained for a 5× super-resolution factor, as shown in Fig. 3a. Since the diffusion reverse process introduces additional noise with a variance of $\tilde{\delta}_t$, different inferences using the same sampling strategy can produce output images with inherent variance. To further mitigate this variability, in addition to these sampling methods, we also explored an averaging strategy to refine the virtual staining quality and suppress the stochasticity at the output pixel super-resolved VS images. Specifically, we performed repeated image inferences using the vanilla diffusion sampling strategy to generate multiple super-resolved VS images for the same field of view (FOV). These images were then averaged on a pixel-wise basis, using 2, 3, and 5 repeats and the resulting averaged virtually stained images are presented in Fig. 3a. We used quantitative metrics to compare these pixel super-resolved VS images generated using different strategies against their corresponding histochemically stained ground truth images, as shown in Fig. 3b. Specifically, these quantitative metrics were calculated using the

virtually stained images from a subset of the testing dataset, consisting of 12 paired image patches (each measuring 960 × 960 pixels) derived from a single patient. This comparative analysis revealed that the mean diffusion sampling strategy achieved the second-highest SSIM scores and the second-lowest LPIPS scores, outperforming both the vanilla and skip diffusion sampling strategies. It is worth noting that both the vanilla sampling and the skip sampling approaches may not surpass the VS performance of cGAN, as illustrated in Fig. 3b. However, by employing the mean sampling and averaging strategy, the diffusion-based virtual staining model can achieve image quality that outperforms that of cGAN. To further highlight the advantages of the mean diffusion sampling strategy, we conducted a paired $t$-test between it and the other strategies; see the "Methods" section and Fig. 3c. The $t$-test results demonstrated that the mean diffusion sampling strategy is statistically superior to the other approaches, except for the vanilla sampling strategy with 5-times averaging. However, if the same number of averaging were to be used for the mean sampling-based diffusion strategy, it would perform superior compared to the vanilla sampling strategy (detailed in the next sub-section).

We also compared in Fig. 3d the VS image inference time per -1 mm² of label-free tissue section for each strategy. Compared to the cGAN-based models, the diffusion-based VS models have a much longer inference time because of the repeated runs of the denoising network during the reverse sampling process—this is a general weakness of the diffusion-based VS models. Since the skip sampling strategy does not require the denoising network inference after the exit point $t_e$, it achieves a decrease in the VS image inference time compared to the vanilla and mean sampling strategies, which is an advantage. Moreover, the inference time of the vanilla sampling strategy with post-sampling averaging can be estimated by multiplying the inference time of the vanilla sampling strategy by the number of averages. Although the five-time averaging strategy in vanilla sampling offers better performance, its significantly longer inference times may limit its applicability in time-sensitive scenarios; in such cases, the mean diffusion sampling strategy should be the preferred method for competitive image inference.

**Evaluation of diffusion-based virtual staining image variance**
In clinical practice, consistent tissue staining quality is crucial for pathologists to make reliable diagnoses. Therefore, the inherent variance in virtually stained images of the same tissue FOV, resulting from different trajectories of the diffusion process, may hinder its clinical applicability. To demonstrate that our diffusion sampling engineering techniques can significantly reduce this variance, we analyzed the coefficient of variation (CV) for virtually stained images generated in different runs of the diffusion reverse process. For this analysis, we used the trained diffusion model for 1× virtual staining (i.e., without downsampling of the input AF microscopy images), and tested the CV performance of three diffusion sampling engineering techniques: mean sampling, mean sampling with 5-times averaging, and vanilla sampling with 5-times averaging. Specifically, we performed each diffusion sampling technique five times, i.e. five independent runs of the mean sampling strategy and 25 runs of both the mean and vanilla sampling strategies. This resulted in five virtually stained images for the same tissue FOV for each approach. The CV map of each method was then calculated by taking the pixel-wise ratio of the standard deviation to the mean for the YCbCr channels of the virtually stained images, as illustrated in Fig. 4a. The virtually stained images appear highly similar to the histochemically stained image, effectively representing lung morphology, including small airways and capillaries. The chroma components (Cb and Cr) reflect the blue and red color information of the virtually stained images, respectively, evaluating the staining quality of H&E, where hematoxylin stains nuclei purplish-blue and eosin stains the extracellular matrix and cytoplasm pink. As shown in Fig. 4a, both of the averaging methods drastically reduced the

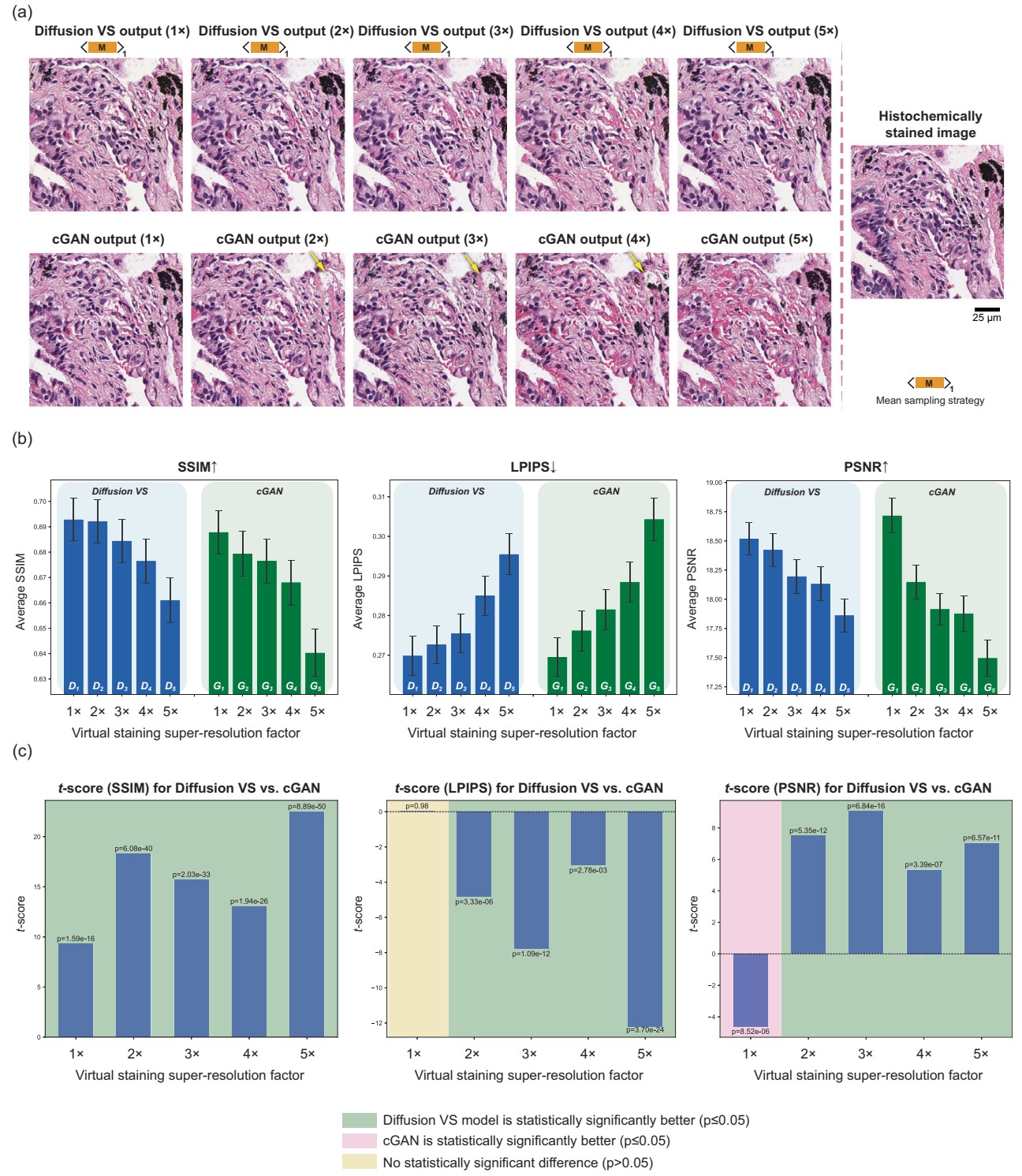

**Fig. 2 | Comparison of super-resolution virtual staining performances of diffusion-based VS models and cGAN-based VS methods. a** Visual comparisons of virtually stained H&E images generated from cGAN-based VS models and our diffusion-based VS models using the mean diffusion sampling strategy. Both models were trained and tested using lower-resolution AF images of unlabeled lung tissue sections, with pixel super-resolution factors ranging from 1× to 5× in each lateral direction. Arrowed regions show failures of the cGAN-based VS model. **b** Bar plots displaying the SSIM, LPIPS, and PSNR metrics averaged across testing virtually stained images for diffusion-based and cGAN-based VS models. These metrics were calculated on $n = 180$ virtually stained and histochemically stained H&E images from 15 blind testing lung samples. The error bars represent the standard error of the mean. $D_x$, $G_x$ denote diffusion-based and cGAN-based VS models, respectively, where $x$ represents the pixel super-resolution factor. **c** Bar plots of $t$-scores calculated through paired two-sided $t$-tests between the diffusion-based VS models and their cGAN-based counterparts with the same super-resolution factor. The green areas show the statistically significantly improved virtual staining performance of the diffusion virtual staining model over the cGAN-based VS model. No adjustments for multiple comparisons were needed. Source data are provided as a Source Data file.

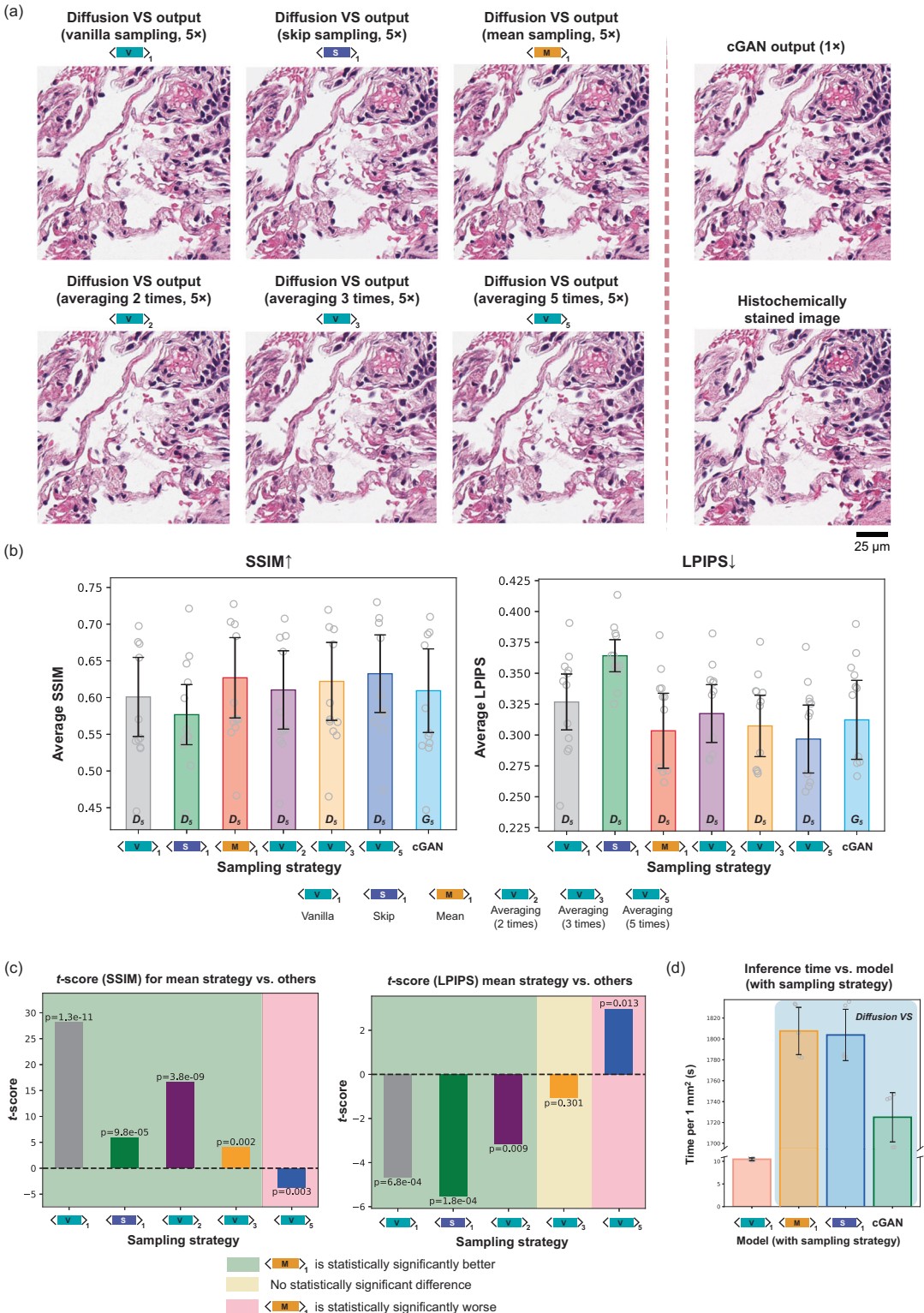

**Fig. 3 | Comparison of performance for different diffusion sampling strategies using the 5× super-resolution diffusion-based VS model. a** Visual comparisons of virtually stained H&E images generated using different sampling strategies with the diffusion-based VS model trained for 5× super-resolution factor. The virtually stained images produced by the cGAN-based VS model (for the 1× case, without super-resolution) and the histochemically stained image of the same FOV are also presented for comparison. An assessment conducted by a certified pathologist (N.P.) revealed strong structural similarity across all image subsegments (e.g., alveoli, blood vessels, and scattered lymphocytes). **b** Bar plots showing the averaged quantitative metrics, including SSIM and LPIPS, comparing the virtually stained images generated from different diffusion sampling strategies shown in (**a**) against their corresponding histochemically stained ground truth images. These metrics were calculated on $n = 12$ FOVs from a blinded testing lung sample. The cGAN results are also displayed for comparison. **c** Bar plots of *t*-scores calculated through paired two-sided *t*-tests between the inference results obtained using the mean diffusion sampling strategy and those from other sampling strategies. The green areas show the statistically significant superiority of the mean diffusion sampling strategy. No adjustments for multiple comparisons were needed. **d** Comparisons of VS image inference time per ~1 mm² of label-free tissue between the cGAN-based VS model and our diffusion-based VS model using three different sampling strategies. The presented time was averaged from 5 independent experiments. The error bars in **b** and **d** represent the standard error of the mean. Source data are provided as a Source Data file.

variance of the diffusion-based image inference across all three channels. Furthermore, with the 5-times averaging approach, the mean diffusion sampling strategy outperformed the vanilla sampling strategy, achieving a mean CV of less than 0.5% for both the Cb and Cr channels. A similar analysis is reported in Fig. 4b with the averaging factors ranging from 2 to 5. The results further confirmed our conclusions that the mean diffusion sampling strategy outperformed the others at all the averaging factors, revealing its superiority over the vanilla sampling strategy by yielding consistent results with smaller variance at its VS images. Note that the reduction in the output variance achieved by averaging with the mean sampling strategy exhibits diminishing returns beyond 5-time averaging; therefore, 5-time averaging was selected as the optimal strategy to showcase our virtual staining performance. Our results demonstrate that a more deterministic diffusion image inference with negligible variations can be achieved in virtual tissue staining using our diffusion sampling process engineering approaches, which is important for practical applications of diffusion-based VS models in digital pathology.

### Generalization to human heart tissue samples

To further demonstrate the robustness and the generalization ability of our diffusion-based VS models on a new type of organ, we employed transfer learning on lung H&E diffusion models ($D_1$ to $D_5$) using a small human heart tissue dataset. This dataset included autofluorescence and histochemically stained image pairs from five heart samples. The resulting heart-specific H&E diffusion models, denoted as $D_x^H$ (where $x$ represents the pixel super-resolution factor, e.g., $D_1^H$ transfer-learned from $D_1$), were subsequently tested on 155 autofluorescence image sets (with each autofluorescence channel having $960 \times 960$ pixels) from 25 unlabeled heart tissue sections not included in the transfer learning process. Our blind testing results reveal that the virtually stained heart H&E images align closely with the histochemically stained ground truth, regardless of the super-resolution factor, as shown in Fig. 5a. The virtually stained heart images provided an accurate representation of cardiac myocytes and the interstitium, clearly visualizing muscle striations and intercalated discs in longitudinal sections, as well as centrally located nuclei in cross sections. This agreement is consistently observed across various FOVs of the heart tissue obtained from different patients. Furthermore, the quantitative metrics presented in Fig. 5b confirm the extended success and consistency of our virtual staining models. Additionally, paired $t$-tests were performed to compare the performance of $D_2^H$ and $D_3^H$ with $D_1^H$. The $p$ values shown in Fig. 5b indicate that these models deliver statistically comparable virtual staining performance for heart samples, even though $D_2^H$ and $D_3^H$ were tested on AF images with lower spatial resolution. These findings highlight the robustness and super-resolution capability of our diffusion-based VS models, reinforcing their potential for accurate and adaptable staining across various tissue and organ types.

## Discussion

The presented success of the Brownian bridge diffusion model for pixel super-resolution virtual staining of label-free tissue, combined with the sampling process engineering, can be readily extended to various existing image reconstruction or enhancement tasks in biomedical imaging. As demonstrated in the "Results" section, the diffusion-based VS models outperformed cGAN-based alternatives in super-resolution virtual staining of label-free tissue images, while the two approaches performed statistically similar for the VS without super-resolution. Diffusion models in general exhibit stable training dynamics[56] and are well-suited to addressing some of the most challenging image reconstruction problems.

One of the major achievements of this work is the development of diffusion sampling process engineering to suppress performance variations in VS of label-free tissue images. A vanilla diffusion model was derived to synthesize images matching the distribution of $x_0$,

where the additional step-wise variance introduced during the reverse process not only stands for an essential component of the diffusion model to match posterior distributions $p_v(x_{t-1}|x_t, y)$ but also enables diversity in the generated images. On the other hand, the consistency of the virtual staining results is crucial for pathologists to make reliable diagnoses. Considering that the profile of $\tilde{\delta}_t$ shows a drastic increase at the end stages of the reverse sampling process (as illustrated in Fig. 1j), using a mean or skip diffusion sampling step that avoids $\tilde{\delta}_t$ in the final sampling steps is important to suppress the stochastic variations in the output of the diffusion model for the same label-free tissue FOV. It is worth noting that this sampling process engineering does not require modifications to the training process or finetuning of a pre-trained model, since the network consistently predicts the error at $x_t$.

We believe that the diffusion sampling process engineering approaches presented in this study for pixel super-resolution virtual tissue staining can be further refined and optimized. Both the mean and skip sampling strategies are constricted by the famous bias-variance tradeoff[57,58]. In other words, the reduction in the variance of a stochastic estimator would inevitably cause an increase of the error bias between the mean of the estimator and the ground truth value. To better understand this trade-off for super-resolution VS of label-free tissue, we optimized the exit point for both the mean and skip diffusion sampling strategies. In a diffusion process consisting of 1000 total steps, we evaluated the performance of the mean and skip sampling strategies across nine different exit points, ranging from 10 to 500. Quantitative image metrics were calculated by comparing the generated VS images to their ground truth counterparts, as shown in Fig. 6. The superior quantitative metrics observed across all exit points $t_e$ further validate the advantages of the mean sampling strategy over the skip sampling strategy. Specifically, for the mean sampling strategy, we observed an increment in LPIPS scores with larger exit points, confirming the necessity of the posterior noise and the bias-variance trade-off. However, the effect of the exit point may differ under different evaluation metrics. The SSIM scores improved at larger exit points, indicating that the addition of the variance term results in a trade-off. Therefore, picking a comprehensive evaluation metric and an appropriate exit point for an optimal result is very important for uses of diffusion-based image inference models in biomedical applications. Furthermore, beyond optimizing the exit points, combining different sampling strategies may also enhance the image inference performance. As demonstrated in Fig. 4a, b, integrating the mean diffusion sampling strategy with subsequent averaging yields superior CV performance compared to other competing strategies. The design of these combined strategies can be more sophisticated; for example, one can apply the averaging strategy to the results inferred from the mean diffusion sampling strategies implemented with different exit points $t_e$. One can also simultaneously apply accelerated sampling strategies, e.g. Denoising Diffusion Implicit Models[59] (DDIM) and Pseudo Linear Multi-Step method[60] (PLMS), with the variance-reduction strategies introduced in this work to achieve faster and better results. These various combinations can be further explored and tailored for different image reconstruction and synthesis applications, beyond the virtual staining of label-free tissue sections.

The diffusion-based VS models ($D_x$) presented in this study are specifically trained for each integer pixel super-resolution factor, making them dedicated models for a particular spatial downsampling factor. To demonstrate the versatility and robustness of our framework, we further explored a universal model ($D_U$) simultaneously trained and tested across all pixel super-resolution factors (from 1× to 5×). The quantitative evaluation of the blind test results for this universal model is illustrated in Supplementary Fig. 2b, c. As expected, the universal model ($D_U$) exhibits a performance trade-off and cannot achieve statistically equivalent performance compared to the dedicated diffusion models ($D_x$), likely due to accommodating diverse

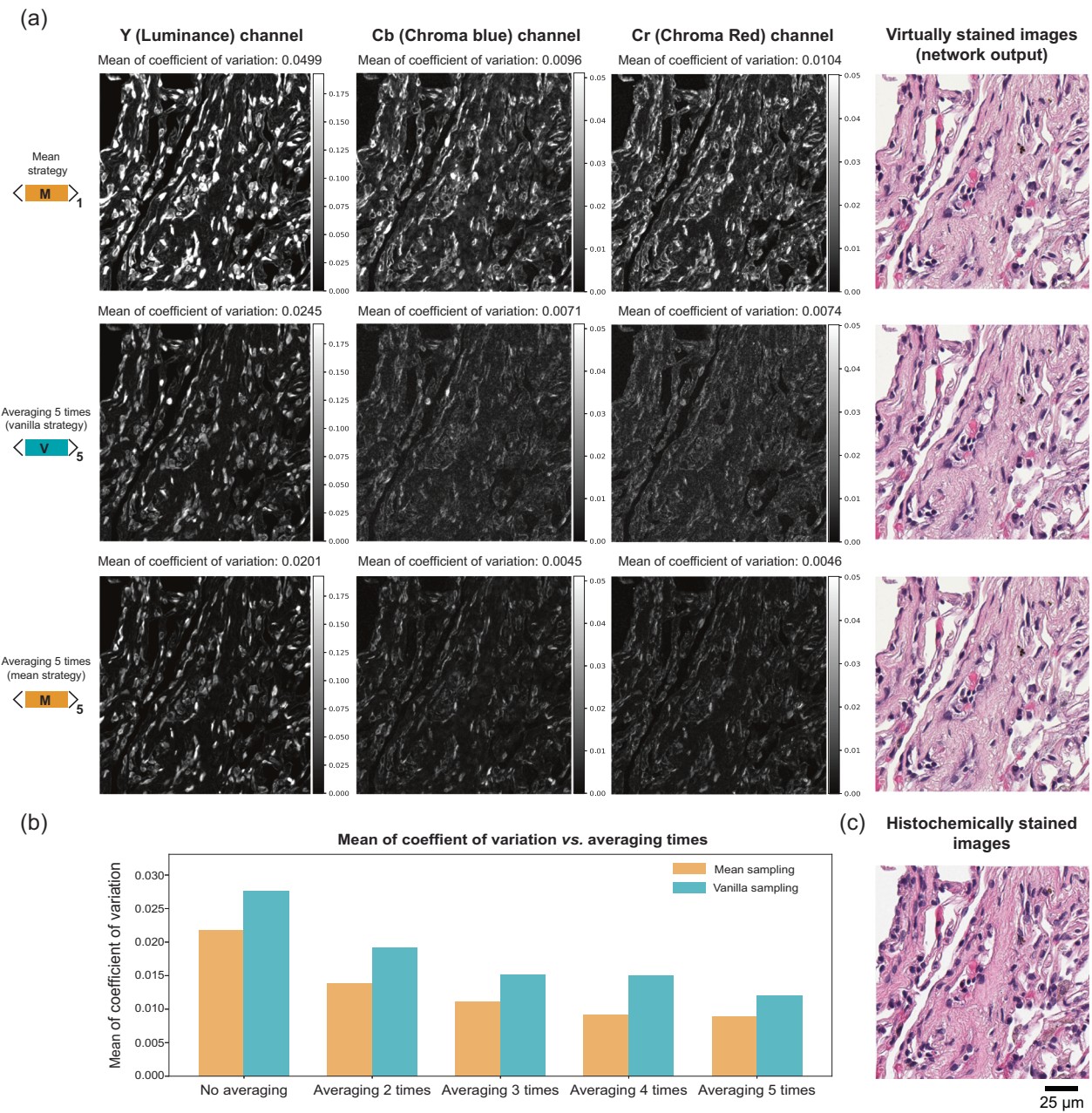

**Fig. 4 | Comparison of the coefficient of variation (CV) for diffusion-based virtually stained images generated in different sampling runs using different diffusion sampling engineering approaches. a** Visualization of the CV maps for the YCbCr channels of the generated virtually stained tissue images, obtained using three different approaches: mean sampling, mean sampling with 5-times averaging, and vanilla sampling with 5-times averaging. The generated virtually stained images of these approaches are also presented in the last column. **b** Plot of the mean CV for mean/skip sampling strategies with different averaging times. The mean CV was calculated across all color channels and pixels of all test image FOVs.
**c** Histochemically stained image of the same FOV in (**a**). Source data are provided as a Source Data file.

super-resolution tasks. Nevertheless, $D_U$ consistently produces high-quality virtually stained images that closely match the corresponding histochemically stained ground truth images across all pixel super-resolution factors. Notably, it successfully reconstructs anthracotic pigment features that the cGAN model failed to capture, as shown in Fig. 2a.

We also investigated the application of our diffusion-based VS models for faster sample scanning by evaluating their performance with a reduced number of input AF channels. Specifically, we trained and tested diffusion VS models using two AF channels (DAPI and TxRed) and three AF channels (DAPI, TxRed, and Cy5) with a 2× super-resolution factor. The visual and quantitative results of these models, termed $D_2^{2ch}$ and $D_2^{3ch}$, are presented in Supplementary Fig. 3. These quantitative results demonstrate that $D_2^{3ch}$, with one AF channel removed, still achieves statistically significant improvements in virtual staining performance compared to the cGAN model ($G_2$) that used all four AF channels. Furthermore, $D_2^{2ch}$, with two AF channels removed, maintains statistically equivalent performance to the cGAN model $G_2$. These results highlight the robustness of our diffusion-based framework and its potential for faster sample scanning by reducing the input image AF channel requirements without compromising performance.

Denoising Diffusion Probabilistic Models[20,21] (DDPM) are among the most widely used and powerful diffusion frameworks for image generation[20,21,24], editing[23,61], and enhancement[49]. Their recent success in

(a)

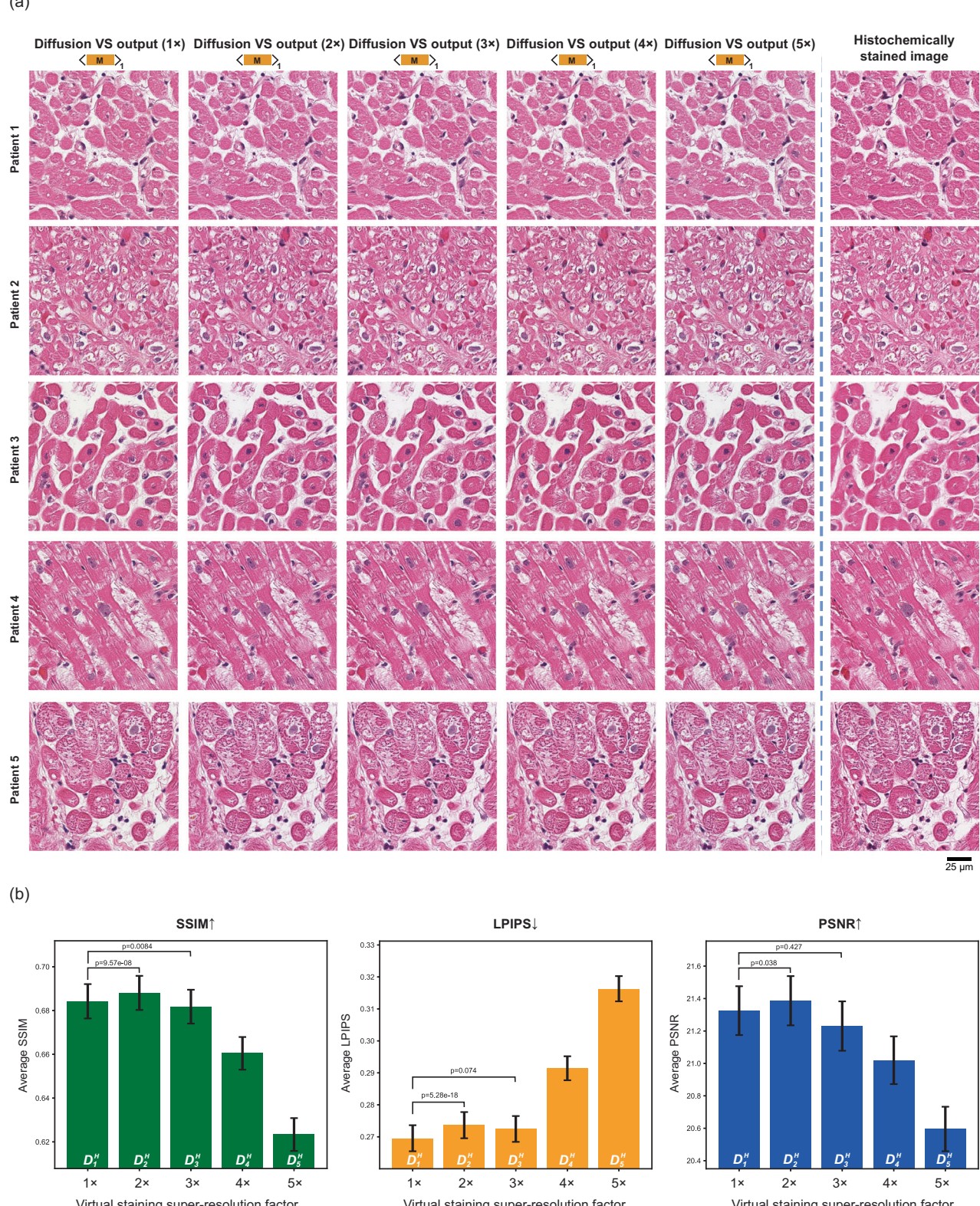

(b)

**Fig. 5 | Super-resolution virtual staining performances on human heart tissue samples using transfer learning. a** Virtually stained H&E images of label-free heart tissue samples, generated by transfer-learned diffusion-based VS models employing the mean sampling strategy. All transfer-learned models (1× to 5×) were trained using five heart tissue sections. **b** Bar plots illustrating the SSIM, LPIPS, and PSNR metrics, averaged across virtually stained testing images for the transfer learned diffusion VS models. These metrics were calculated by comparing $n = 155$ virtually stained images to their corresponding histochemically stained H&E images from 25 unseen, unlabeled human heart tissue samples. Error bars indicate the standard error of the mean. $D_x^H$ represents the diffusion-based virtual heart H&E model transfer learned from the $D_x$ (virtual lung H&E model), where the $x$ represents the pixel super-resolution factor. The $p$ values are calculated through paired two-sided $t$-tests between the metrics of models with different pixel super resolution factors. No adjustments for multiple comparisons were needed. Source data are provided as a Source Data file.

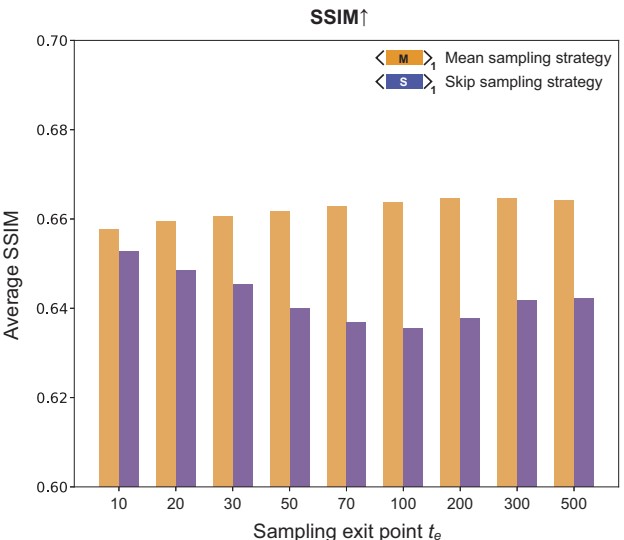
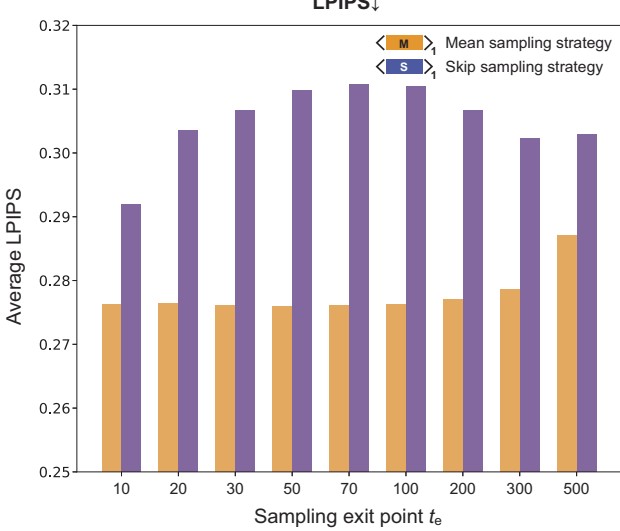

**Fig. 6 | Optimization of the sampling exit point $t_e$ for both the mean and skip diffusion sampling strategies.** Bar plots display the average values of the SSIM and LPIPS metrics for image inference results obtained using the mean and skip diffusion sampling strategies configured with different sampling exit points $t_e$. This exit point optimization was performed using the diffusion-based VS model for the 1× case. Source data are provided as a Source Data file.

stain transformation related tasks[29,30] highlights their potential in computational pathology. To further demonstrate the effectiveness of our diffusion-based VS models ($D_x$), we compared their performance against DDPM-based VS models ($P_x$), as shown in Supplementary Fig. 4. To provide a fair comparison, the DDPM-based models employed the original DDPM sampling process[20] and used the same denoising U-Net architecture as our models. Quantitative evaluations on blind test data, presented in Supplementary Fig. 4b, c, clearly show that our models outperform the DDPM-based virtual staining counterparts, underscoring the effectiveness of our proposed approach.

Although we demonstrated the efficacy of our technique through virtual H&E staining of human lung and heart tissues, our label-free approach can be generalized to other histochemical stains and a variety of organ systems. This adaptability is reinforced by prior successes with different virtual staining methods[1]. Furthermore, recent advances in virtual staining of microscopic images from cancerous samples[9,62] underscore the potential applicability of our method to cancerous tissues, representing a promising avenue for future research. Such studies would further validate and enhance the clinical relevance and diagnostic utility of pixel super-resolution virtual staining methods.

In conclusion, we introduced a diffusion-based model for pixel super-resolution virtual staining of label-free, lower resolution autofluorescence microscopy images, demonstrating its superiority over the state-of-the-art virtual tissue staining approaches. Furthermore, we developed several diffusion sampling engineering techniques, which not only improved the image quality of virtually stained images but also resulted in more consistent image inference with a lower statistical variation, which is crucial for the wide-spread uses of these AI-based image reconstruction/synthesis approaches in biomedical settings.

## Methods
### Sample preparation, image acquisition, and histochemical H&E staining

Lung and heart tissue samples for this research were sourced from existing, de-identified tissue blocks acquired by UCLA Translational Pathology Core Laboratory (TPCL) with informed consent from all patients. The study complied with the ethical standards of UCLA Institutional Review Board (IRB) with authorization under IRB approval

# 18-001029. The study involved lung specimens from 33 individual patients and heart samples from 30 individual patients. For each patient, a tissue section approximately 4 μm thick was sliced from the unlabeled tissue blocks, deparaffinized, and mounted on glass slides. Autofluorescence images of the lung/heart tissue sections were acquired using a Leica DMI8 microscope equipped with a 40×/0.95 NA objective lens (Leica HC PL APO 40×/0.95 DRY), controlled by the Leica LAS X software for automated microscopy. The autofluorescence images of label-free tissue sections were captured under four distinct fluorescence filter cubes: DAPI (Semrock OSFI3-DAPI-5060C, EX 377/50 nm, EM 447/60 nm), TxRed (Semrock OSFI3-TXRED-4040C, EX 562/40 nm, EM 624/40 nm), FITC (Semrock FITC-2024B-OFX, EX 485/20 nm, EM 522/24 nm), and Cy5 (Semrock CY5-4040C-OFX, EX 628/40 nm, EM 692/40 nm). Images were captured using a scientific complementary metal-oxide-semiconductor (sCMOS) sensor (Leica DFC 9000 GTC), with an exposure time of around 300 ms for all four autofluorescence channels. After performing autofluorescence imaging, the same unlabeled tissue sections were sent to UCLA TPCL for standard histochemical H&E staining (used as ground truth). The stained tissue slides were then scanned and digitized using a bright-field slide scanner (Leica Biosystems Aperio AT2).

### Dataset division and preparation

The training and testing dataset comprised paired autofluorescence images and their corresponding bright-field histochemically stained H&E images. For the lung experiments, 1051 paired autofluorescence-H&E microscopic image patches (each with 2048 × 2048 pixels) from 18 patients were used for training, and 180 paired image patches (each with 960 × 960 pixels) were reserved for blind testing, obtained from 15 de-identified patients not included in the training set. As for the heart experiments, 163 paired autofluorescence-H&E images (each with 2048 × 2048 pixels) from 5 patients were used for transfer learning, and 155 image pairs (each with 960 × 960 pixels) from 25 patients were used for blind testing. During each training epoch, the paired image FOVs were subdivided into smaller 192 × 192-pixel patches, normalized for zero mean and unit variance, and further augmented through random flipping and rotation to ensure robust model training.

In the image registration workflow, a rigid registration approach was initially applied at the whole slide image (WSI) level. The maximum

cross-correlation coefficient was calculated for each WSI pair, allowing for the estimation of the rotation angles and shifts. This facilitated the spatial alignment of the histochemically stained WSIs to their auto-fluorescence counterparts. Following this, the slides underwent a finer registration at the image patch level. The WSIs were segmented into smaller FOV pairs of 3248 × 3248 pixels (~528 × 528 μm²), followed by a multi-modal affine image registration algorithm to adjust for shifts, sizing differences, and rotations between the histology and auto-fluorescence image FOVs. Before cropping WSIs into smaller local FOVs, an intensity normalization of the autofluorescence channels was performed. In the final phase, small local paired FOVs were cropped to 2048 × 2048 pixels (~333 × 333 μm²) to reduce edge artifacts and underwent an iterative elastic pyramid cross-correlation registration[3,63,64] to achieve pixel-level alignment. During this elastic registration process, an initial virtual staining network was trained to match the style of the autofluorescence images to the style of the brightfield H&E images. The resulting roughly-stained images and their histochemically stained counterparts were fed into the elastic pyramid registration algorithm to obtain transformation maps which were then applied to correct the local discrepancies in the ground truth images to better align with their corresponding autofluorescence images. These training and registration cycles were repeated until precise pixel-level registration was achieved. As a final step, the manual data cleaning was employed to remove images with obvious artifacts like tissue tearing or image blurring in out-of-focus areas.

## Baseline VS models using cGAN

All baseline VS models, which were compared with our diffusion-based VS models, were built on a state-of-the-art structurally-conditioned GAN architecture[2,3,5,6,9,13,14,51]. The generator network employs a five-level U-Net structure, while the discriminator network is a convolutional neural network-based classifier. The training loss for the generator includes adversarial loss and pixel-based structural loss terms, such as mean absolute error (MAE) loss and total variation (TV) loss[51]. Meanwhile, the training loss for the discriminator utilizes a least squares loss function, following ref. 51. The generator and the discriminator networks were updated at a frequency ratio of 3:1. The learning rate for optimizing the generator network was set at $1 \times 10^{-4}$, while for the discriminator network, it was set at $1 \times 10^{-5}$. The Adam[65] optimizer was employed for network training, and the batch size was set as 8 for all the model training.

## Brownian bridge diffusion process

We utilized the Brownian bridge diffusion process to model the conditional diffusion and apply it to transform the lower resolution autofluorescence images of label-free tissue $y_0 \in \mathbb{R}^{\frac{H}{N} \times \frac{W}{N} \times 4}$ (where $N$ is the pixel super-resolution factor) into the target histochemically stained higher resolution brightfield images $x_0 \in \mathbb{R}^{H \times W \times 3}$. Since the diffusion sampling process requires the conditional and target images to have identical dimensions, the autofluorescence images of label-free tissue $y_0$ were first processed to match the dimensions of the target image domain through a shallow convolutional neural network (CNN), as depicted in Fig. 7a. This shallow network is composed of two convolutional layers and a pixel-shuffle[66] layer for dimension adaptation, denoted as:

$$y = f_c(y_0) \tag{1}$$

where $y \in \mathbb{R}^{H \times W \times 3}$ has a dimension matched with the histochemically stained ground truth images $x_0$. The forward Brownian bridge with an initial state $x_0$ and terminal state $y$ is defined as:

$$q(x_t|x_0, x_T = y) = \mathcal{N}\left(\left(1 - \frac{t}{T}\right)x_0 + \frac{t}{T}y, \frac{2t(T-t)}{T^2}I\right) \tag{2}$$

where $T$ is the total sampling steps, and $t$ is the intermediate time step. $T$ was set to 1000 during both the forward and reverse processes. By denoting $m_t = \frac{t}{T}$ and $\delta_t = \frac{2t(T-t)}{T^2}$, we can reparametrize the distribution of $x_t$ as:

$$x_t = x_0 + m_t(y - x_0) + \sqrt{\delta_t}\epsilon_t, \ \epsilon_t \sim N(0, I) \tag{3}$$

Equation (3) shows that the arbitrary intermediate step $\boldsymbol{x_t}$ can be directly sampled using $x_0$ and $y$ during the forward diffusion process, represented as "direct sampling" in Fig. 1b. The shallow network output $y$ is concatenated with the noisy image $\boldsymbol{x_t}$ and fed into a denoising network $\epsilon_\theta$. This denoising network is trained to estimate $\boldsymbol{x_0}$ from $\boldsymbol{x_t}$ and $t$. In other words, it is trained to estimate $m_t(y - x_0) + \sqrt{\delta_t}\epsilon_t$. This denoising network $\epsilon_\theta$ is designed based on a U-Net[67] structure and consists of one down-sampling path and one up-sampling path with skip connections in-between the two paths. Figure 7b illustrates the architecture of the U-Net, where each path has four consecutive levels and each level contains a residual block and an attention block. The adjacent levels in the down-sampling and up-sampling path are connected with a 2 × 2 average pooling and 2 × 2 nearest interpolation, respectively. The middle block of the U-Net is the concatenation of two residual blocks and one attention block. The attention block adopts a multi-head attention mechanism[68]. The timestep $t$ is embedded through a linear layer with SiLU[69,70] pre-layer activation and added to the input features of the residual block.

The loss function of $\epsilon_\theta$ is defined as:

$$L = \sum_t \gamma_t E_{(x_0, y), \epsilon_t} ||m_t(y - x_0) + \sqrt{\delta_t}\epsilon_t - \epsilon_\theta(x_t, t)||_2^2 \tag{4}$$

where $\gamma_t$ is the weight for each $t$. During the training, a uniform sampling schedule was utilized, setting $\gamma_t = 1$ for all the sampling steps, $t$.

The vanilla reverse process can be shown to be a Gaussian process with mean $\mu'_t(x_t, y)$ and variance $\tilde{\delta}_t I$

$$p_v(x_{t-1}|x_t, y) = \mathcal{N}\left(x_{t-1}; \mu'_t(x_t, y), \tilde{\delta}_t I\right) \tag{5}$$

$$\mu'_t(x_t, y) = c_{xt}x_t + c_{yt}y - c_{et}\epsilon_\theta(x_t, t) \tag{6}$$

$$\tilde{\delta}_t = \frac{\delta_{t|t-1}\delta_{t-1}}{\delta_t} \tag{7}$$

The expressions of $c_{xt}, c_{yt}, c_{et}$, and $\delta_{t|t-1}$ and the details of Eqs. (4) and (6) are detailed in Supplementary Note 1.

During the blind testing, a test autofluorescence image $y_0$ is first processed by $f_c$ to generate the image $x_T = y$. From this, the image at time $T - 1$, $x_{T-1}$, can be estimated using Eqs. (5)–(6), i.e., $x_{T-1} = c_{xT}x_T + c_{yT}y - c_{eT}\epsilon_\theta(x_T, T) + \sqrt{\tilde{\delta}_T}z$, where $z \sim N(0, I)$.

This step is repeated for $T$ iterations to estimate the target virtual stained image $x_0$, as denoted with "sample step by step" in Fig. 1c–e. This process is the vanilla sampling strategy illustrated in Fig. 1c, g. This strategy can be represented as a Markov chain with learned Gaussian transitions, starting with $p(x_T) = \mathcal{N}(x_T; y, 0)$:

$$p_\theta^{\text{vanilla}} = p(x_T)\prod_{t=1}^T p_v(x_{t-1}|x_t, y) \tag{8}$$

The mean sampling strategy, as depicted in Fig. 1d, h, go through the same vanilla sampling steps when $t_e < t < T$, where $t_e$ is defined as the exit point in reverse sampling process. For $t \leq t_e$, this strategy

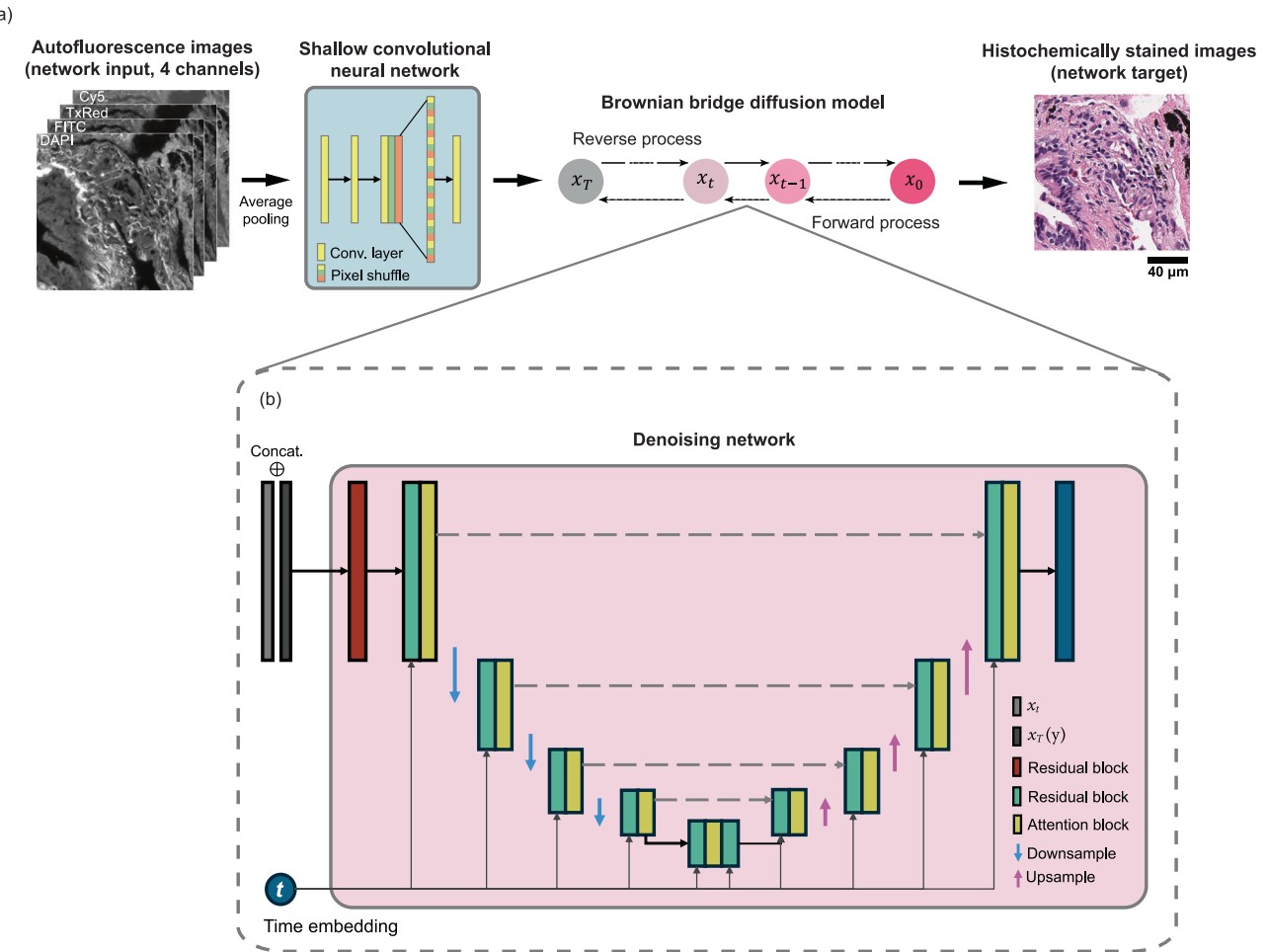

**Fig. 7 | Network architecture for the diffusion-based pixel super-resolution virtual staining model. a** The pipeline of the forward and reverse sampling processes. The detailed architecture of the shallow convolutional neural network used for dimension matching is also presented. **b** Detailed architecture of the denoising network used at each step of both the forward and reverse sampling processes.

estimates $x_{t-1}$ from $x_t$ using a mean sampling step as follows:

$$p_m(x_{t-1}|x_t,y) = \mathcal{N}\left(x_{t-1}; \mu'_t(x_t,y), 0\right) \qquad (9)$$

where the $x_{t-1}$ is directly estimated from $\mu'_t(x_t,y)$, without adding additional variance pattern $\widetilde{\delta}_t$ in the sampling step, i.e., $x_{t-1} = c_{xt}x_t + c_{yt}y - c_{\epsilon t}\epsilon_\theta(x_t,t)$ for $t \le t_e$. Therefore, the mean sampling strategy can be formulated as:

$$p_\theta^{mean} = p(x_T)\prod_{t=1}^{t_e}p_m(x_{t-1}|x_t,y)\prod_{t=t_e}^{T}p_\nu(x_{t-1}|x_t,y) \qquad (8)$$

Similarly, the skip sampling strategy follows the same vanilla sampling steps until the exit point $t = t_e$:

$$p_\theta^{skip} = p(x_T)\prod_{t=t_e}^{T}p_\nu(x_{t-1}|x_t,y) \qquad (9)$$

However, it diverges by directly estimating the final virtual stained image from the state $x_{t_e}$, as illustrated in Fig. 1e, i, i.e.:

$$x_0^{skip} = x_{t_e} - \epsilon_\theta\left(x_{t_e}, t_e\right) \qquad (10)$$

During the training, the shallow convolution neural network and the denoising network were all optimized using the AdamW optimizer[71], starting with a learning rate of $1 \times 10^{-4}$. A batch size of 16

was maintained throughout the training phase, and the network converged after approximately 72 h of training. During the blind testing, we empirically set $t_e = 50$.

## Quantitative image evaluation metrics

To quantitatively evaluate the performance of H&E virtual staining, we utilized standard metrics of SSIM and LPIPS. The SSIM is defined as:

$$SSIM(a,b) = \frac{(2\mu_a\mu_b + C_1)(2\sigma_{a,b} + C_2)}{(\mu_a^2 + \mu_b^2 + C_1)(\sigma_a^2 + \sigma_b^2 + C_2)} \qquad (11)$$

where $\mu_a$ and $\mu_b$ are the mean values of $a$ and $b$, which represent the two images being compared. $\sigma_a$ and $\sigma_b$ are the standard deviations of $a$ and $b$. $\sigma_{a,b}$ is the cross-covariance of $a$ and $b$. $C_1$ and $C_2$ are constants that are used to avoid division by zero.

For LPIPS metrics, we used a pretrained VGG model to evaluate the learned perceptual similarity between the generated virtually stained images $m$ and their corresponding histochemically stained images $m_0$. These image pairs were fed into the pretrained VGG network and their feature stack from $L$ layers were extracted as $\hat{n}^l, \hat{n}_0^l \in \mathbb{R}^{H_l \times W_l \times C_l}$ for layer $l$. The LPIPS score was calculated as:

$$d(m,m_0) = \sum_l \frac{1}{H_l W_l}\sum_{h,w}\left\|\left(\hat{n}_{hw}^l - \hat{n}_{0hw}^l\right)\right\|_2^2 \qquad (12)$$

The PSNR is defined as:

$$\text{PSNR} = 10\log_{10}\left(\frac{\max(A)^2}{\text{MSE}}\right) \qquad (13)$$

where $A$ represents the histochemically stained brightfield H&E images and max($A$) is the maximum pixel value of the image $A$. The mean squared error (MSE) is defined as:

$$\text{MSE} = \frac{1}{MN}\sum_{m}^{M}\sum_{n}^{N}\left[A_{mn} - B_{mn}\right]^2 \qquad (14)$$

where $B$ represents the virtually stained H&E images. $m$, $n$ are the pixel indices, and $MN$ denotes the total number of pixels in each image.

To perform the spatial frequency spectrum analysis, the raw autofluorescence DAPI image was first bilinearly upsampled from its original size of $\frac{960}{x}\times\frac{960}{x}$ pixels ($x$ is the spatial undersampling factor) to $960 \times 960$ pixels, matching the dimensions of the grayscale virtually stained and histochemically stained images. A two-dimensional Fourier Transform was then applied to the upsampled autofluorescence DAPI image, as well as the virtually stained and the histochemically stained image. For consistency, both the virtual and histochemical H&E images were processed in grayscale for this analysis. The radially averaged power spectrum was calculated following the method reported in Wang et al.[64].

## Statistical significance analysis

In Fig. 2, paired two-sided $t$-tests were used to assess whether our diffusion-based VS models showed statistically significant difference from the virtual staining performance of cGAN-based VS models. These tests were performed across 180 unique FOVs, using the SSIM and LPIPS metrics calculated for both our diffusion-based VS model and the corresponding cGAN-based VS counterpart for the same super-resolution factor. The null hypothesis for the paired two-sided $t$-test assumes that the two models, given the same super-resolution factor, should have identical means for the same FOV. We used a statistical significance level of 0.05 to reject the null hypothesis in favor of the alternative, i.e., the two-sided $t$-test assumed 2.5% probability level to each side (superiority and inferiority). Note that a lower (higher) score for LPIPS (SSIM) metrics is desired for improved VS performance; therefore our results in Fig. 2 revealed a statistically significant improvement of our diffusion-based VS models over their cGAN-based counterparts for LPIPS when $t<0, p\le0.05$ and for SSIM when $t>0, p\le0.05$. Similarly, as illustrated in Fig. 3, paired two-sided $t$-tests were performed using the SSIM and LPIPS metrics to compare the virtual staining performance of the mean sampling VS strategy against other diffusion-based VS strategies. Same as before, for both LPIPS (when $t<0$) and SSIM (when $t>0$), a $p$ value of $\le0.05$ from the two-sided $t$ test revealed the statistical superiority of the mean sampling-based diffusion strategy over the other VS approaches, showing inferiority only to the vanilla sampling strategy with 5-times averaging.

## Other implementation details

All image preprocessing and registration were performed using MATLAB version R2022b. All network training and testing tasks were conducted on a desktop computer equipped with an Intel Core i9-13900K CPU, 64 GB of memory, and an NVIDIA GeForce RTX 4090 GPU. The code for training the diffusion models was developed in Python 3.9.19 using PyTorch 2.2.1.

## Reporting summary

Further information on research design is available in the Nature Portfolio Reporting Summary linked to this article.

## Data availability

The authors declare that all data supporting the results of this study are available within the main text and Supplementary Information. Whole tissue slides were obtained under UCLA IRB #18-001029 from UCLA Health. Example testing images and network models are available together with our code at: https://github.com/Yijie-Zhang/Super-resolved-virtual-staining. Source data are provided with this paper.

## Code availability

Deep learning models reported in this work used standard libraries and scripts that are publicly available in PyTorch. The codes used to develop the model, perform the analyses and generate results in this study is publicly available and has been deposited in GitHub at https://github.com/Yijie-Zhang/Super-resolved-virtual-staining under Apache-2.0 license. The specified version of the code associated with this publication is archived in Zenodo and is accessible via https://doi.org/10.5281/zenodo.15226375 (ref. 72).

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

## Acknowledgements

The authors acknowledge the funding of NSF Biophotonics Program (A.O.) and the NIH National Center for Interventional Biophotonic Technologies (P41—A.O.).

## Author contributions

A.O. conceived the research, Y.L. and N.P. imaged the unlabeled tissue sections, Y.L. and Y.Z. developed the image processing pipeline and prepared the dataset, Y.Z., L.H., Y.L. and H.C. trained the neural networks. Y.Z., L.H. and N.P. performed the result analysis and statistical study. Y.Z. and L.H. prepared the manuscript, and all authors contributed to the manuscript. A.O. supervised the research.

## Competing interests

A.O. is the founder of a company (Pictor Labs) that commercializes virtual staining technology. A.O., Y.Z. and L.H. have a pending patent application on diffusion model-based virtual staining of label-free tissue. The remaining authors declare no competing interests.
