## [Transparent Peer Review file · Nature Communications]

Pixel super-resolved virtual staining of label-free tissue using diffusion models

Corresponding Author: Dr Aydogan Ozcan

Version 0:

Reviewer comments:

Reviewer #1

(Remarks to the Author)

In this work, the authors report a Brownian bridge diffusion model that achieves both virtual H&E staining and super-resolution reconstruction of lower resolution, label-free autofluorescence images. Considerable attention is additionally given to diffusion model sampling engineering techniques in attempt to mitigate well-known variability inherent to diffusion models during the reverse process sampling. This is a strength of this study, since as discussed, this will be critical for applying diffusion models for virtual staining in biomedical applications, where consistent inference from the same label-free input is necessary. In my opinion, this work is technically sound, well-organized and reasonably validated. This work is appropriate for Nature Communications, and I can support its publication following some minor revisions. The following are my specific comments:

1. The super-resolution image restoration capabilities have been demonstrated with high-quality results on images which have undergone pixel binning to artificially reduce the space bandwidth product. Has the model performance been investigated on a more realistic use case where the label-free autofluorescence images are collected using, for example, a lower NA microscope to facilitate faster scanning or an optical system degraded by aberrations? I do not think demonstrating this is strictly necessary for publication, but this could significantly enhance the impact of the work. While it appears that currently the network had to be re-trained for each integer spatial downsampling factor, would it be possible to train the model to restore arbitrary (within a reasonable range) factors of resolution degradation?
2. Do other standard image comparison metrics such as PSNR or PCC indicate similar performance trends to SSIM and LPIPS?
3. In the example images shown in Fig. 2a, it is easy to appreciate the stain quality and morphological similarity achieved by the diffusion model as compared to the ground truth histochemically-stained image. However, it is more challenging to evaluate the super-resolution capability of the model in restoring resolution to match the true H&E image, at least beyond a gross impression. The authors should consider showing some frequency domain analysis of example images or similar method to provide further qualitative or quantitative evidence of the resolution restoring capability of the model.
4. In Fig. 1 d-e, I think x_t is meant to be x_{t_e} indicating the exit point for the mean and skip diffusion sampling methods.
5. In Fig. 1, the notation in the legend and diagrams indicating direct sampling, and sampling step-by-step is currently unclear. Further explanation or re-working of this would be helpful.
6. In multiple results figures, the caption should indicate the meaning of the error bars, i.e. standard deviation or standard error of the mean, etc.
7. Do the authors have an explanation for why the inference time shown in Fig. 3d is greater for the mean sampling strategy compared to the vanilla approach without averaging?
8. In comparing performance to the cGAN approach, only results for the mean sampling strategy are shown. Can the authors comment on the results for the other sampling strategies compared to cGAN?
9. The reported Brownian bridge diffusion model appears to offer superior virtual staining performance compared to previous cGAN work in many aspects. Could this approach be expected to confer any additional performance advantages such as virtual staining of lower SNR images to allow faster slide scanning, equivalent performance with fewer autofluorescence channels, or extensibility to slide-free imaging modalities instead of unstained slides? While supervised training remains a challenge for that use case, the simultaneous virtual staining and image restoration capability could prove valuable.

(Remarks on code availability)

Reviewer #2

(Remarks to the Author)

This paper proposes a diffusion-based virtual staining model to enhance the resolution and consistency of autofluorescence-guided histological imaging, aiming to provide an alternative to traditional H&E staining. The study evaluates the model's performance using a dataset of paired autofluorescence and bright-field histologically stained images. The authors implement three diffusion sampling strategies and compare them against a cGAN-based baseline. The results are presented through metrics like the coefficient of variation and visualised with maps and statistical analyses. While the proposed method shows promise in generating high-resolution virtually stained images, I found some limitations:

1. The introduction relies heavily on preprints. While they can be valuable, the lack of peer-reviewed citations raises questions about the reliability and maturity of the referenced work.
2. The test set is really small, just 12 images from one patient. That's not enough to get a clear picture of how well the method will work on other data or patients.
3. It doesn't look like there's a separate validation set to tune the model or check for overfitting during training. That's a bit of a gap.
4. Since the test images are from only one patient, it's hard to know if the model works well across different tissues or conditions.
5. The baseline comparison is self-implemented, which might unintentionally favour the proposed method. It would be more convincing if pre-existing, well-tuned cGAN implementations were used.
6. Training on 192×192 patches is practical, but small patches can lose important global context. It's unclear if the method addresses this limitation, like by overlapping patches or using multi-scale inputs.
7. The choice of 5-times averaging is mentioned but not explained. Why five? Is there a specific reason for this number, or was it chosen arbitrarily? A brief justification would strengthen the methodology.
8. Regarding computational costs, sampling strategies like 5-times averaging might add extra computational overhead, but there's no discussion of how practical this is for larger datasets or real-world use.
9. There's no mention of a pathologist or domain expert reviewing the virtually stained outputs to confirm they're biologically meaningful.

Based on the strengths of the paper, including its novel use of diffusion models for virtual staining and the demonstrated improvements over a cGAN baseline, I believe this work has significant potential to contribute to the field of histological imaging. However, the current manuscript has several limitations that I listed before. Therefore, I recommend publication pending revisions.

(Remarks on code availability)

I can see the code and the provided instructions for running it. However, running the code requires logging into the system, which I prefer not to do at this stage of the review to maintain my anonymity.

Version 1:

Reviewer comments:

Reviewer #1

(Remarks to the Author)

In this revision, the authors have made considerable efforts to improve the manuscript and have sufficiently addressed the reviewer comments. In particular, the universal methods capable of restoration of arbitrary input resolution, and the interpretation of these results are a significant and interesting addition to the paper. I fully support acceptance of the revised manuscript for publication.

(Remarks on code availability)

Reviewer #3

(Remarks to the Author)

A major issue of this study is the lack of cancerous samples in the experiments. Based on the figures, most of samples used do not contain tumor tissues, and virtual staining on normal nuclei is relatively easy. Morphologically, the cancerous cell is characterized by a large nucleus, having an irregular size and shape, the nucleoli are prominent, the cytoplasm is scarce

and intensely colored or, on the contrary, is pale.

As the motivation of the study is to produce virtually stained samples for further AI training or clinical diagnosis, verification on staining of cancerous samples of different kinds of cancers is vital. This should include specimens of major types of cancer, including breast cancer, lung cancer, cervical cancer and colorectal cancer, lymphoma.

To demonstrate whether the proposed virtual staining approach could properly stain the tumorous tissues of the main kinds of cancer should be provided. The test set is too small for evaluation.

In evaluation, the authors should provide quantitative comparison of the methods with the reference standards (real HE stained with pathologists' annotations on the tumor regions) with respect to the Jaccard similarity coefficient. This is to demonstrate how well individual virtual staining approaches perform to stain tumour tissues / samples of various kinds of cancer.

Moreover, the authors should also perform experiments to show that with the outputs of the proposed method, how much improvements could be made using some SOTA AI models?

My suggestion is similar to the second reviewer that there's no quantification evaluation for pathologists reviewing the virtually stained outputs to confirm they're biologically meaningful. More importantly, the major issue of this study is the lack of cancerous samples in the experiments. To demonstrate whether the proposed virtual staining approach could properly stain the tumorous tissues of the main kinds of cancer should be provided. The test set is too small for evaluation.

Therefore, I recommend publication pending revisions.

(Remarks on code availability)

Please provide clear instructions on the Github web page for all training, testing and evaluation process. So far, training and evaluation codes are missing, which will be hard for reproducibility.

Version 2:

Reviewer comments:

Reviewer #3

(Remarks to the Author)

The authors have addressed most comments. One minor suggestion is that please use box-plots instead of error bars. In addition, please provide p-values for statistical analysis results.

(Remarks on code availability)

We sincerely thank the referees for their reviews and the constructive feedback that we have received on our manuscript "***Super-resolved virtual staining of label-free tissue using diffusion models***" submitted to *Nature Communications* (Manuscript ID: NCOMMS-24-68310).

As detailed below, we have revised our manuscript in response to the reviewers' comments. The original referee comments are shown in black color, whereas for ease of communication, our answers are provided in blue. Our revisions have also been marked in the main text and supplementary information files using yellow highlighting.

Summary of our revisions:

- **Expansion of the test dataset and generalization to heart tissue samples.** In this revision, we have significantly expanded the test dataset to 180 unique fields-of-view (FOVs) from 15 de-identified patients, providing a large dataset for robust quantitative and statistical analysis. Furthermore, beyond the lung tissue sample, we have demonstrated the success of our diffusion-based super-resolved virtual staining technique on a new organ — the heart. The heart models were trained using transfer learning with data from only 5 patients and evaluated on an additional set of 178 FOVs from 25 de-identified patients. To showcase these results, we have added the following figures and paragraphs:
 - **Revised Figure 2.** Comparison of super-resolution virtual staining performances of diffusion-based VS models and cGAN-based VS methods.
 - **Newly added Figure 5.** Super-resolution virtual staining performances on human heart tissue samples using transfer learning.
 - **Newly added Subsection "Generalization to human heart tissue samples"** in the Results section.
- **Added comprehensive quantitative comparisons between super-resolved virtually stained images and their histochemically stained counterparts.** In addition to the previous SSIM and LPIPS-based quality metric analysis, we included a new quantitative metric, i.e., PSNR. Moreover, a spatial frequency-based analysis and the calculation of power spectrum cross-sections were added to our revised manuscript to quantitatively demonstrate the super-resolution capability of the diffusion-based VS models. For these quantitative analyses, we have modified and added the following figures and paragraphs to our revised manuscript:
 - **Newly added Supplementary Figure 1.** Spatial frequency spectrum analysis of virtually stained images generated by diffusion-based VS models.
 - **Newly added last paragraph in the subsection "Super-resolved virtual staining of unlabeled tissue sections using a diffusion model"** in the Results section.
 - **Revised Figure 2.** Comparison of super-resolution virtual staining performances of diffusion-based VS models and cGAN-based VS methods.
- **Added visual and quantitative comparison results between separately trained dedicated diffusion VS models for different super-resolution factors (as previously demonstrated in our work) and a universal VS model capable of handling arbitrary super-resolution factors.** The results are presented in the newly added Supplementary Figure 2 and explained in a newly added 4th paragraph in the Discussion section as listed below:
 - **Newly added 4th paragraph in the Discussion section**
 - **Newly added Supplementary Figure 2.** Comparison of super-resolution virtual staining performances of dedicated and universal diffusion-based VS models.
- **Demonstration of the success of our framework with fewer auto-fluorescence channels for faster scanning of tissue samples.** We have added the following figures and paragraphs:
 - **Newly added Supplementary Figure 3.** Evaluation of super-resolution virtual staining performances of diffusion-based VS models using a reduced number of input autofluorescence channels.
 - **Newly added 5th paragraph in the Discussion section**

List of figure changes:

Revised:

- **Figure 1.** Diffusion model-based super-resolution virtual staining of unlabeled tissue sections.
- **Figure 2.** Comparison of super-resolution virtual staining performances of diffusion-based VS models and cGAN-based VS methods
- **Figure 3.** Comparison of performance for different diffusion sampling strategies using the 5x super-resolution diffusion-based VS model

Newly added:

- **Figure 5.** Super-resolution virtual staining performances on human heart tissue samples using transfer learning.
- **Supplementary Figure 1.** Spatial frequency spectrum analysis of virtually stained images generated by diffusion-based VS models.
- **Supplementary Figure 2.** Comparison of super-resolution virtual staining performances of dedicated and universal diffusion-based VS models.
- **Supplementary Figure 3.** Evaluation of super-resolution virtual staining performances of diffusion-based VS models using a reduced number of input autofluorescence channels.

Reviewer #1:

In this work, the authors report a Brownian bridge diffusion model that achieves both virtual H&E staining and super-resolution reconstruction of lower resolution, label-free autofluorescence images. Considerable attention is additionally given to diffusion model sampling engineering techniques in attempt to mitigate well-known variability inherent to diffusion models during the reverse process sampling. This is a strength of this study, since as discussed, this will be critical for applying diffusion models for virtual staining in biomedical applications, where consistent inference from the same label-free input is necessary. In my opinion, this work is technically sound, well-organized and reasonably validated. This work is appropriate for Nature Communications, and I can support its publication following some minor revisions. The following are my specific comments:

We thank the reviewer for the positive feedback and the constructive comments that helped us further enhance our manuscript. All the specific comments are addressed below:

- (1) The super-resolution image restoration capabilities have been demonstrated with high-quality results on images which have undergone pixel binning to artificially reduce the space bandwidth product. Has the model performance been investigated on a more realistic use case where the label-free autofluorescence images are collected using, for example, a lower NA microscope to facilitate faster scanning or an optical system degraded by aberrations? I do not think demonstrating this is strictly necessary for publication, but this could significantly enhance the impact of the work. While it appears that currently the network had to be re-trained for each integer spatial downsampling factor, would it be possible to train the model to restore arbitrary (within a reasonable range) factors of resolution degradation?

We thank the reviewer for these thoughtful comments. Although we used pixel binning to reduce the space bandwidth product, the demonstrated image reconstruction and virtual staining performance under such information loss highlight the robustness of our diffusion-based VS models. Regarding the possibility of training the model for arbitrary super-resolution factors, our diffusion-based virtual staining model indeed supports this flexibility. Following the referee's suggestion, we demonstrated this capability by training a universal model (D_U) using inputs randomly undersampled by factors ranging from $1\times$ to $5\times$. We then tested this universal model alongside the previously presented separately trained models (D_1 to D_5 , where D_x corresponds to the model trained on inputs with an undersampling factor of x) on 180 unique image fields-of-view from 15 unseen lung tissue sections. The visual and quantitative comparison results of this new analysis are presented in the **new Supplementary Figure 2**. The quantitative results and statistical t -test analyses indicate that the universal model underperforms compared to the separately trained diffusion-based VS models. This represents a trade-off, as the universal model provides the flexibility to handle diverse

factors of resolution degradation. However, despite this relative trade-off, the universal model consistently generates high-quality virtually stained images that have a close match to the corresponding histochemically stained ground truth images across all super-resolution factors. For example, the universal model successfully reconstructed anthracotic pigment features that the cGAN models failed to capture, as shown in Fig. 2(a).

To present the versatility and success of this universal model and clarify its trade-off on super-resolution virtual staining performance, we have added the following paragraph in the Discussion section of our revised manuscript:

"...The diffusion-based VS models (D_x) presented in this study are specifically trained for each integer super-resolution factor, making them dedicated models for a particular spatial downsampling factor. To demonstrate the versatility and robustness of our framework, we further explored a universal model (D_U) simultaneously trained and tested across all super-resolution factors (from 1x to 5x). The quantitative evaluation of the blind test results for this universal model is illustrated in Supplementary Fig. 2(b-c). As expected, the universal model (D_U) exhibits a performance trade-off and cannot achieve statistically equivalent performance compared to the dedicated diffusion models (D_x), likely due to accommodating diverse super-resolution tasks. Nevertheless, D_U consistently produces high-quality virtually stained images that closely match the corresponding histochemically stained ground truth images across all super-resolution factors. Notably, it successfully reconstructs anthracotic pigment features that the cGAN model failed to capture, as shown in Fig. 2(a)."

- (2) Do other standard image comparison metrics such as PSNR or PCC indicate similar performance trends to SSIM and LPIPS?

We thank the reviewer for this insightful suggestion. In response, we have added two new quantitative performance metrics in our comparisons: (1) we have added a PSNR analysis to compare the performance of our diffusion-based virtual staining models with those of cGAN models, as shown in the revised Fig. 2; and (2) a spatial frequency-based analysis and the calculation of power spectrum cross-sections were added to our revised manuscript to quantitatively demonstrate the super-resolution capability of the diffusion-based VS models. For these quantitative analyses, we have modified and added the following figures and paragraphs to our revised manuscript:

- o Newly added Supplementary Figure 1. Spatial frequency spectrum analysis of virtually stained images generated by diffusion-based VS models.
- o Newly added last paragraph in the subsection "Super-resolved virtual staining of unlabeled tissue sections using a diffusion model" in the Results section.
- o Revised Figure 2. Comparison of super-resolution virtual staining performances of diffusion-based VS models and cGAN-based VS methods.

To ensure a more comprehensive evaluation, we expanded the testing dataset to 180 image FOVs obtained from 15 de-identified patients. The quantitative results presented in the revised Fig. 2(b), along with statistical t -test results, demonstrate that our diffusion-based virtual staining model achieves statistically significant improvements in virtual staining performance. The PSNR metric exhibits the same trend as SSIM and LPIPS metrics for this larger dataset, reinforcing the consistency of our findings.

We have added the following sentences to the revised Results section:

"...During the blind testing phase, both approaches (diffusion vs. cGAN) were applied to 180 autofluorescence image sets (with each autofluorescence channel having 960x960 pixels) from 15 unlabeled lung tissue sections that were never used during training to generate VS images at various super-resolution factors, as shown in Fig. 2(a)."

We have also added the calculation details of PSNR into the Methods section:

"The PSNR is defined as:

$$PSNR = 10 \log_{10} \left(\frac{\max(A)^2}{MSE} \right) \quad (13)$$

where A represents the histochemically stained brightfield H&E images and $\max(A)$ is the maximum pixel value of the image A . The mean squared error (MSE) can be denoted as:

$$MSE = \frac{1}{MN} \sum_m^M \sum_n^N [A_{mn} - B_{mn}]^2 \quad (14)$$

where B represents the virtually stained H&E images. m, n are the pixel indices, and MN denotes the total number of pixels in each image..."

To further evaluate the generalizability of our model, we have also extended our analysis to heart tissue sections representing a different organ. We applied transfer learning using only 5 heart tissue sections and conducted blind testing on 178 additional FOVs from 25 de-identified patients. The results of PSNR, SSIM, and LPIPS analyses for these tests are presented in **the newly added Fig. 5**. The successful performance on heart samples, along with the consistent trends across all three metrics in Fig. 5, underscore the generalizability of our model to different tissue types and organs.

To showcase these results, we have added a new subsection titled: "Generalization to human heart tissue samples" to the Results section:

"...To further demonstrate the robustness and the generalization ability of our diffusion-based VS models on a new type of organ, we employed transfer learning on lung H&E diffusion models (D_1 to D_5) using a small human heart tissue dataset. This dataset included autofluorescence and histochemically stained image pairs from five heart samples. The resulting heart-specific H&E diffusion models, denoted as D_x^H (where x represents the super-resolution factor, e.g., D_1^H transfer-learned from D_1), were subsequently tested on 178 autofluorescence image sets (with each autofluorescence channel having 960×960 pixels) from 25 unlabeled heart tissue sections not included in the transfer learning process. Our blind testing results reveal that the virtually stained heart H&E images align closely with the histochemically stained ground truth, regardless of the super-resolution factor, as shown in Fig. 5(a). This agreement is consistently observed across various FOVs of the heart tissue obtained from different patients. Furthermore, the quantitative metrics presented in Fig. 5(b) confirm the extended success and consistency of our virtual staining models. Additionally, paired t-tests were performed to compare the performance of D_2^H and D_3^H with D_1^H . The p-values shown in Fig. 5(b) indicate that these models deliver statistically comparable virtual staining performance for heart samples, even though D_2^H and D_3^H were tested on AF images with lower spatial resolution. These findings highlight the robustness and super-resolution capability of our diffusion-based VS models, reinforcing their potential for accurate and adaptable staining across various tissue and organ types."

Our spatial frequency-based analyses will be summarized in response to the next question of the referee.

- (3) In the example images shown in Fig. 2a, it is easy to appreciate the stain quality and morphological similarity achieved by the diffusion model as compared to the ground truth histochemically-stained image. However, it is more challenging to evaluate the super-resolution capability of the model in restoring resolution to match the true H&E image, at least beyond a gross impression. The authors should consider showing some frequency domain analysis of example images or similar method to provide further qualitative or quantitative evidence of the resolution restoring capability of the model.

Following the referee's suggestion and to better illustrate the super-resolution capability of our diffusion-based VS models, we conducted a spatial frequency spectrum analysis. This analysis compared the spatial frequency information of the input autofluorescence DAPI images, the virtually stained images, and the corresponding histochemically stained images. The cross-sections of the radially averaged power spectra of these images, presented in **Supplementary Figure 1(b)**, demonstrate a significant enhancement in the spatial frequencies of the virtually stained images compared to the input DAPI images. This result provides robust evidence of our model's ability to restore resolution effectively, aligning it closely with the ground truth H&E-stained images.

To showcase these analyses, we have also included the following text in the revised Results section:

"...To highlight the super-resolution capabilities of the diffusion-based VS model, a spatial frequency spectrum analysis was performed on the input autofluorescence DAPI images, the virtually stained images generated by the diffusion model, and their corresponding histochemically stained ground truth images for all super-resolution factors. The results are presented in Supplementary Fig. 1(a); also refer to the Methods section for details. The cross-sections of the radially averaged power spectra^{49,50}, as shown in Supplementary Fig. 1(b), reveal a significant enhancement in the spatial frequency spectra of the VS images compared to the lower-resolution autofluorescence inputs. Notably, the spectra of the VS images have a good alignment with those of the histochemically stained ground truth, underscoring the diffusion-based VS model's ability to enhance spatial resolution."

We have also added the implementation details of the spatial frequency spectrum analysis to the revised Methods section:

"...To perform the spatial frequency spectrum analysis, the raw autofluorescence DAPI image was first bilinearly upsampled from its original size of $\frac{960}{x} \times \frac{960}{x}$ pixels (x is the spatial undersampling factor) to 960x960 pixels, matching the dimensions of the grayscale virtually stained and histochemically stained images. A two-dimensional Fourier Transform was then applied to the upsampled autofluorescence DAPI image, as well as the virtually stained and the histochemically stained image. For consistency, both the virtual and histochemical H&E images were processed in grayscale for this analysis. The radially averaged power spectrum was calculated following the method reported in Wang et al⁶⁷."

- (4) In Fig. 1 d-e, I think x_t is meant to be x_{t_e} indicating the exit point for the mean and skip diffusion sampling methods.

We have accordingly corrected the labels in the revised Figure 1.

- (5) In Fig. 1, the notation in the legend and diagrams indicating direct sampling, and sampling step-by-step is currently unclear. Further explanation or re-working of this would be helpful.

We thank the reviewer for the comment. To clarify the terms used in Fig. 1, we have added the following explanations to the Methods section:

"...Eq. (3) shows that the arbitrary intermediate step x_t can be directly sampled using x_0 and y during the forward diffusion process, represented as "direct sampling" in Fig. 1(b)."

"...This step is repeated for T iterations to estimate the target virtual stained image x_0 , as denoted with "sample step by step" in Fig.1(c-e)."

- (6) In multiple results figures, the caption should indicate the meaning of the error bars, i.e. standard deviation or standard error of the mean, etc.

We thank the reviewer for this valuable suggestion. All the error bars in the figures represent the standard error of the mean. We have updated the captions of all relevant figures to explicitly clarify the meaning of the error bars.

- (7) Do the authors have an explanation for why the inference time shown in Fig. 3d is greater for the mean sampling strategy compared to the vanilla approach without averaging?

Theoretically, the mean sampling strategy and the vanilla approach should exhibit similar inference times, as both require running the denoising network approximately 1000 times during the reverse process, which dominates the computation. Additionally, the mean sampling strategy does not involve sampling or adding variance after the exit point, suggesting a slightly smaller inference time compared to the vanilla approach.

The observed time difference in Fig. 3(d) may be attributed to GPU inference jitter, potentially caused by variations in GPU temperature or hardware-specific differences. To ensure a more robust and accurate comparison, we conducted five consecutive inferences for each model on the same GPU. We updated Fig. 3(d) to include error bars representing the standard error of the mean. As shown in the revised Fig. 3(d), the inference time for the mean sampling strategy is slightly smaller than for the vanilla approach, aligning with our expectations.

- (8) In comparing performance to the cGAN approach, only results for the mean sampling strategy are shown. Can the authors comment on the results for the other sampling strategies compared to cGAN?

We thank the reviewer for the suggestion. In our study, the mean sampling strategy was identified as the optimal approach for achieving superior virtual staining performance. When considering output variance, the 5-time averaging strategy can be combined with either the vanilla or mean sampling strategies to provide enhanced inference. Consequently, we focused on the comparison of cGAN with the optimal sampling strategy, omitting comparisons with suboptimal methods.

To provide a more comprehensive analysis, we have revised Fig. 3(b) by including the quantitative results of cGAN in the last bar plot column as a baseline. The updated results indicate that suboptimal methods, such as the skip sampling strategy, do not surpass the virtual staining performance of cGAN.

We have also added the following explanation to the revised Results section:

"...It is worth noting that both the vanilla sampling and the skip sampling approaches may not surpass the VS performance of cGAN, as illustrated in Fig. 3(b). However, by employing the mean sampling and averaging strategy, the diffusion-based virtual staining model can achieve image quality that outperforms that of cGAN."

- (9) The reported Brownian bridge diffusion model appears to offer superior virtual staining performance compared to previous cGAN work in many aspects. Could this approach be expected to confer any additional performance advantages such as virtual staining of lower SNR images to allow faster slide scanning, equivalent performance with fewer autofluorescence channels, or extensibility to slide-free imaging modalities instead of unstained slides? While supervised training remains a challenge for that use case, the simultaneous virtual staining and image restoration capability could prove valuable.

We thank the reviewer for this valuable suggestion. Our diffusion-based virtual staining model indeed demonstrates robustness with fewer autofluorescence channels, which is advantageous for faster scanning of a sample. To validate this, we trained our diffusion-based virtual staining model using 2-channel (D_2^{2ch} , DAPI and TxRed) and 3-channel (D_2^{3ch} , DAPI, TxRed, and Cy5) autofluorescence inputs at a super-resolution factor of $2\times$. These models were then tested alongside the originally presented 4-channel diffusion model (D_2) and the corresponding cGAN model (G_2) on 180 test FOVs from 15 lung tissue sections.

The visual and quantitative results, presented in the newly added **Supplementary Figure 3**, indicate that even with one or two fewer channels, our models (D_2^{2ch} , D_2^{3ch}) achieve statistically significant performance improvements or equivalence compared to the cGAN model (G_2) that uses 4 autofluorescence image channels. This demonstrates the ability of our approach to achieve high-quality results with reduced input information, underscoring its potential for faster scanning of tissue samples.

To showcase these new results, we have also included the following paragraph in the Discussion section:

*"...We also investigated the application of our diffusion-based VS models for faster sample scanning by evaluating their performance with a reduced number of input AF channels. Specifically, we trained and tested diffusion VS models using two AF channels (DAPI and TxRed) and three AF channels (DAPI, TxRed, and Cy5) with a $2\times$ super-resolution factor. The visual and quantitative results of these models, termed D_2^{2ch} and D_2^{3ch} , are presented in **Supplementary Fig. 3**. These quantitative results demonstrate that D_2^{3ch} , with one AF channel removed, still achieves statistically significant improvements in virtual staining performance*

compared to the cGAN model (G_2) that used all four AF channels. Furthermore, D_2^{2ch} , with two AF channels removed, maintains statistically equivalent performance to the cGAN model G_2 . These results highlight the robustness of our diffusion-based framework and its potential for faster sample scanning by reducing the input image AF channel requirements without compromising performance."

Reviewer #2:

This paper proposes a diffusion-based virtual staining model to enhance the resolution and consistency of autofluorescence-guided histological imaging, aiming to provide an alternative to traditional H&E staining. The study evaluates the model's performance using a dataset of paired autofluorescence and bright-field histologically stained images. The authors implement three diffusion sampling strategies and compare them against a cGAN-based baseline. The results are presented through metrics like the coefficient of variation and visualised with maps and statistical analyses. While the proposed method shows promise in generating high-resolution virtually stained images, I found some limitations:

We thank the reviewer for their positive feedback and suggestions, which have helped us to further enhance our manuscript and improve its clarity. Point-by-point responses are included below.

- (1) The introduction relies heavily on preprints. While they can be valuable, the lack of peer-reviewed citations raises questions about the reliability and maturity of the referenced work.

We appreciate the reviewer's observation and have taken steps to enhance the reliability and maturity of the referenced work in the Introduction section. Specifically:

- a. For preprint papers that have since been published, we updated the citations to their published (peer-reviewed) versions.
- b. We have replaced some of the unpublished preprints with alternative, peer-reviewed works that align closely with the context of our study.
- c. We have retained a small number of preprint references due to their significance in the field.

Following these adjustments, the number of preprint references in the Introduction section has been reduced from 11 to 3.

- (2) The test set is really small, just 12 images from one patient. That's not enough to get a clear picture of how well the method will work on other data or patients.

We thank the reviewer for this valuable comment. To address this concern and demonstrate the generalizability of our model, we tested our diffusion-based virtual staining models and the corresponding cGAN models on a significantly larger dataset consisting of **180 image fields-of-view (FOVs) from 15 de-identified patients**.

For a more comprehensive analysis, we also included an additional quantitative metric, PSNR, in the evaluation. The quantitative results and statistical t -test analyses for this expanded test dataset are presented in the **revised Fig. 2**. These results consistently confirm that our diffusion-based virtual staining model outperforms the corresponding cGAN models across various super-resolution factors.

We have also added the following sentence to the revised Results section:

"...During the blind testing phase, both approaches (diffusion vs. cGAN) were applied to 180 autofluorescence image sets (with each autofluorescence channel having 960x960 pixels) from 15 unlabeled lung tissue sections that were never used during training to generate VS images at various super-resolution factors, as shown in Fig. 2(a)."

Additionally, we conducted spatial frequency analysis to quantitatively demonstrate the improvement in spatial resolution of the VS images generated by diffusion models compared to their corresponding low-resolution AF inputs. The results are summarized in the **newly added Supplementary Figure 2**.

To further demonstrate the model's ability to generalize to different tissues or organs, we conducted a transfer learning experiment. **Specifically, we fine-tuned the lung diffusion-based VS models using only 5 heart tissue sections and then tested the transfer-learned models on 178 image FOVs from 25 de-identified patients.** The visual and quantitative results, shown in the new Fig. 5, demonstrate the successful application of the transfer-learned models to heart tissue samples, illustrating the model's ability to generalize to a new organ.

We have also added a new subsection "**Generalization to human heart tissue samples**" to the Results section to elaborate on the success of the transfer-learned model on heart samples:

*"...To further demonstrate the robustness and the generalization ability of our diffusion-based VS models on a new type of organ, we employed transfer learning on lung H&E diffusion models (D_1 to D_5) using a small human heart tissue dataset. This dataset included autofluorescence and histochemically stained image pairs from five heart samples. The resulting heart-specific H&E diffusion models, denoted as D_x^H (where x represents the super-resolution factor, e.g., D_1^H transfer-learned from D_1), were subsequently tested on 178 autofluorescence image sets (with each autofluorescence channel having 960×960 pixels) from 25 unlabeled heart tissue sections not included in the transfer learning process. **Our blind testing results reveal that the virtually stained heart H&E images align closely with the histochemically stained ground truth, regardless of the super-resolution factor, as shown in Fig. 5(a).** This agreement is consistently observed across various FOVs of the heart tissue obtained from different patients. Furthermore, the quantitative metrics presented in Fig. 5(b) confirm the extended success and consistency of our virtual staining models. Additionally, paired t -tests were performed to compare the performance of D_2^H and D_3^H with D_1^H . **The p -values shown in Fig. 5(b) indicate that these models deliver statistically comparable virtual staining performance for heart samples, even though D_2^H and D_3^H were tested on AF images with lower spatial resolution.** These findings highlight the robustness and super-resolution capability of our diffusion-based VS models, reinforcing their potential for accurate and adaptable staining across various tissue and organ types."*

We believe this extended analysis, covering 40 (15+25) de-identified patients, robustly demonstrates the consistent success and applicability of our model across different tissue types and organs from different patients.

- (3) It doesn't look like there's a separate validation set to tune the model or check for overfitting during training. That's a bit of a gap.

We evaluated model checkpoints using a validation set of 50 input-target image pairs (each with 192×192 pixels) that were entirely separate from both the training and testing datasets. Specifically, for the $3\times$ model, we assessed all checkpoints and identified the one corresponding to approximately 15 training epochs as optimal. Then, we consistently used the checkpoint corresponding to ~ 15 epochs across all training models for inference.

To further ensure robustness and mitigate concerns about overfitting, we incorporated additional analyses. As detailed earlier, we have evaluated the model's performance on an expanded lung tissue dataset and conducted a transfer learning experiment for another organ (heart). The successful virtual staining performance on testing FOVs from 40 de-identified patients in total (15 for lung and 25 for heart) confirms the model's ability to generalize effectively to new data and different tissues, providing strong evidence against overfitting.

- (4) Since the test images are from only one patient, it's hard to know if the model works well across different tissues or conditions.

We thank the reviewer for raising this concern. To address this issue, we have extended our evaluation to include a significantly larger dataset with 180 FOVs from 15 de-identified patients. This analysis, detailed in the revised manuscript and Fig. 2, demonstrates the consistent performance of our diffusion-based virtual staining model across diverse patient samples, confirming its generalizability.

Additionally, to assess the model's robustness across different tissue types, we conducted a transfer learning experiment on heart tissue sections. The transfer-learned models, fine-tuned using only 5 heart tissue sections, were tested on 178 FOVs from 25 de-identified patients. As shown in the new Fig. 5, the models successfully generalized to this new tissue type, further illustrating their adaptability and robustness under varying conditions.

- (5) The baseline comparison is self-implemented, which might unintentionally favour the proposed method. It would be more convincing if pre-existing, well-tuned cGAN implementations were used.

We sincerely thank the reviewer for pointing this out. However, we would like to emphasize that the self-implemented cGAN baseline models used in our work already delivered competitive virtual staining performance, establishing them as strong baseline models. We conducted a comprehensive review of the published literature to identify suitable, pre-existing, well-tuned cGAN baseline models. However, we could not locate a virtual staining model specifically designed for lung samples that maps 4 autofluorescence channels (DAPI, TxRed, FITC, and Cy5) into the bright-field H&E domain. The most comparable model we identified is a published virtual staining model that uses 2 autofluorescence channels (DAPI and TxRed) as input for transformation into the lung H&E domain, as referenced in ref [*].

Therefore, we evaluated the performance of this published model of ref [*] against our self-implemented cGAN model under a 1× super-resolution factor. The results, presented in the figure shown below, reveal that the virtually stained images generated by the published model are significantly inferior to those produced by our self-implemented cGAN model, both visually and quantitatively, as assessed by PSNR, SSIM, and LPIPS metrics. Furthermore, even after transfer learning the published model using the same training dataset utilized in our study, its performance remained significantly worse than that of our self-implemented model, as demonstrated visually and quantitatively using the same evaluation metrics (PSNR, SSIM, and LPIPS); also see the figure in the following page for detail of this comparison.

* Li, Y., Pillar, N., Li, J. et al. Virtual histological staining of unlabeled autopsy tissue. Nat Commun 15, 1684 (2024). <https://doi.org/10.1038/s41467-024-46077-2>

* Li, Y., Pillar, N., Li, J. *et al.* Virtual histological staining of unlabeled autopsy tissue. *Nat Commun* 15, 1684 (2024). <https://doi.org/10.1038/s41467-024-46077-2>

Figure caption: Comparison of virtual H&E staining performance between the published model (Y.L. *et al.*'s model) before and after transfer learning and the self-implemented cGAN model, against ground truth images obtained from standard histochemical staining. (a) Visual comparisons across four individual fields of view (FOVs). (b-d) Quantitative image quality evaluations and statistical analyses using the metrics of (b) PSNR, (c) SSIM, and (d) LPIPS.

(6) Training on 192×192 patches is practical, but small patches can lose important global context. It's unclear if the method addresses this limitation, like by overlapping patches or using multi-scale inputs.

We want to emphasize that we **employ patch-wise virtual staining and super-resolution, followed by patch stitching to generate larger virtually stained FOVs and whole slide-level images – which has**

been the norm in various virtual staining efforts so far. And the choice of 192×192 image patches in our method is guided by two main considerations:

a. Training and inference efficiency. Training and inference efficiency often require balancing computational resources with model capacity. Many diffusion-based generative models adopt a two-stage approach involving a computationally intensive diffusion step followed by an upscaling step, where lower-resolution images generated in the first stage are adapted to higher resolutions. This strategy has consistently demonstrated efficacy and efficiency across various generative tasks and image domains, as shown in former publications:

Podell, D., Rombach, R., Esser, P., Ommer, B. SDXL: Improving latent diffusion models for high-resolution image synthesis. The Twelfth International Conference on Learning Representations (ICLR) (2024).

Pelykh, A., Mercanoglu Sincan, O., Bowden, R. Giving a hand to diffusion models: a two-stage approach to improving conditional human image generation. arXiv preprint arXiv:2403.10731 (2024).

Fukuda, T., Chen, Q. A two-stage method with diffusion models for single-image view synthesis. Proceedings of the 9th International Conference on Multimedia Systems and Signal Processing (ICMSSP) (2024).

→ Our work aligns with this paradigm by employing patch-wise virtual staining and super-resolution, followed by patch stitching to generate larger virtually stained FOVs.

b. Patch size and task-specific requirements: The selected patch size of 192×192 pixels corresponds to approximately 30×30 μm², which is comparable to previous works in the field, such as:

Li, Y., Pillar, N., Li, J. et al. Virtual histological staining of unlabeled autopsy tissue. Nat Commun 15, 1684 (2024). ~40×40 μm²

Yang, X., Bai, B., Zhang, Y., et al. Virtual birefringence imaging and histological staining of amyloid deposits in label-free tissue using autofluorescence microscopy and deep learning. Nat Commun 15, 7978 (2024). ~40×40 μm²

The success of pathological evaluations and diagnoses in these former studies on virtually stained whole slide images of human tissue samples also supports our patch size selection.

(7) The choice of 5-times averaging is mentioned but not explained. Why five? Is there a specific reason for this number, or was it chosen arbitrarily? A brief justification would strengthen the methodology.

We thank the reviewer for pointing this out. As shown in Fig.4(b), the reduction of the output variance by averaging with the mean sampling strategy has a small improvement beyond 5-time averaging. As the reviewer also mentioned in comment # (8) below, more averaging introduces an extra computational overhead. Considering the trade-off between performance and computation time, 5-times averaging was selected as the optimal averaging strategy. Following the referee's comment, we have added the following sentences to our revised Results section to further expand on this:

"...Note that the reduction in the output variance achieved by averaging with the mean sampling strategy exhibits diminishing returns beyond 5-time averaging; therefore, 5-time averaging was selected as the optimal strategy to showcase our virtual staining performance."

(8) Regarding computational costs, sampling strategies like 5-times averaging might add extra computational overhead, but there's no discussion of how practical this is for larger datasets or real-world use.

We appreciate the reviewer for raising this important point. For digital pathology applications, virtual staining inference time is not a bottleneck since the time and cost-savings achieved by eliminating human histotechnologist time and laborious and toxic chemical staining protocols are major factors in favor of virtual staining. Therefore, the virtual staining-related performance improvements achieved by our diffusion-based models are meaningful, and the inference time of our models does not constitute a challenge for practical

settings in digital pathology workflow. It is indeed true that the demonstrated mean- and skip-sampling strategies would increase the inference overhead compared to previous methods, such as cGAN-based virtual staining, thus potentially demanding more computational resources for deployment in clinical settings. However, we also would like to point out that our presented methods do bring considerable performance improvements over cGAN, as illustrated and quantified in our Results section, and such an improvement is not achievable by cGAN even though the same computational resource is available. In fact, test-time scaling has been gradually accepted as a more effective measure than scaling up the model size/capacity to improve performance on hard test cases, as evidenced by a series of former studies in machine learning literature such as:

Huang, G. Dynamic neural networks: advantages and challenges. *National Science Review* 11, 8 (2024).

Snell, C., Lee, J., Xu, K., Kumar, A. Scaling LLM test-time compute optimally can be more effective than scaling model parameters. *arXiv preprint arXiv:2408.03314* (2024).

Ravishankar, R., Patel, Z., Rajasegaran, J., Malik, J. Scaling properties of diffusion models for perceptual tasks. *arXiv preprint arXiv:2411.08034* (2024).

Besides, our method, as a diffusion model, provides flexibility to inference depending on the hardware's computing power. For example, naive acceleration practices including reducing the total number of sampling steps and increasing the step size, along with other advanced approaches introduced in Refs. [51, 52] can be applied to alleviate the demand for computational resources of our method. The flexibility of our method, as well as possible acceleration strategies, are discussed in the Discussion section, cited below:

"...The design of these combined strategies can be more sophisticated; for example, one can apply the averaging strategy to the results inferred from the mean diffusion sampling strategies implemented with different exit points t_e . One can also simultaneously apply accelerated sampling strategies, e.g. Denoising Diffusion Implicit Models⁵¹ (DDIM) and Pseudo Linear Multi-Step method⁵² (PLMS), with the variance-reduction strategies introduced in this work to achieve faster and better results. These various combinations can be further explored and tailored for different image reconstruction and synthesis applications, beyond the virtual staining of label-free tissue sections."

- (9) There's no mention of a pathologist or domain expert reviewing the virtually stained outputs to confirm they're biologically meaningful

We would like to point out that analyses on pathologically-meaningful features were conducted by a board-certified pathologist (a co-author of our manuscript, Dr. Pillar); see for example the Results section, cited below:

"...Specifically, as indicated in the arrowed region of Fig. 2(a), due to spatial resolution loss of the AF microscopy images of label-free tissue samples, cGAN-based models failed to reconstruct stained regions of anthracotic pigment, which are important in lung pathology for disease diagnosis. In contrast, our diffusion-based super-resolution VS models consistently stained these black pigments, presenting a good match to the histochemically stained images."

Remarks on code:

I can see the code and the provided instructions for running it. However, running the code requires logging into the system, which I prefer not to do at this stage of the review to maintain my anonymity.

We thank the reviewer for the remarks. We have uploaded the codes to GitHub for public release. We have also updated the code availability statement in the main text.

We sincerely thank the referees for their reviews and the constructive feedback that we have received on our manuscript "**Pixel super-resolved virtual staining of label-free tissue using diffusion models**" submitted to *Nature Communications* (Manuscript ID: NCOMMS-24-68310A).

As detailed below, we have revised our manuscript in response to the reviewers' comments. The original referee comments are shown in black color, whereas for ease of communication, our answers are provided in blue. Our revisions have also been marked in the main text and supplementary information files using yellow highlighting.

Summary of our 2nd round of revisions:

- **Explanation of the potential extension and future applications of our approach to cancerous tissue:**
 - **Newly added text in the Discussion section**
- **Included additional pathological evaluations comparing the virtually stained images with their histochemically stained counterparts, conducted by a board-certified pathologist.** These assessments are presented in newly added text within the Results section and figure captions.
- **Included additional visual and quantitative comparison results between our diffusion virtual staining models and other state-of-the-art models, including the Denoising Diffusion Probabilistic Models (DDPM).** The results are presented in the newly added **Supplementary Figure 4** and explained in the revised Discussion section as listed below:
 - **Newly added 6th paragraph in the Discussion section**
 - **Newly added Supplementary Figure 4.** Comparative evaluation of pixel super-resolution virtual staining performance of our diffusion VS models against DDPM-based models.
- **Clarified our terminology by changing "super-resolution" to "pixel super-resolution" to avoid confusion with nanoscopy techniques that surpass the diffraction limit of light.** This change is reflected in the updated paper title and throughout the main text. Additionally, we revised the third paragraph in the Introduction section to provide further clarification.

List of figure changes:

Newly added: Supplementary Figure 4. Comparative evaluation of pixel super-resolution virtual staining performance of our diffusion VS models against DDPM-based models.

Reviewer #1:

In this revision, the authors have made considerable efforts to improve the manuscript and have sufficiently addressed the reviewer comments. In particular, the universal methods capable of restoration of arbitrary input resolution, and the interpretation of these results are a significant and interesting addition to the paper. I fully support acceptance of the revised manuscript for publication.

We sincerely thank the reviewer for the positive feedback and valuable suggestions throughout the review process. These insightful suggestions have significantly strengthened our work, and we deeply appreciate your recommendation for acceptance.

Reviewer #3:

- (1) A major issue of this study is the lack of cancerous samples in the experiments. Based on the figures, most of samples used do not contain tumor tissues, and virtual staining on normal nuclei is relatively easy. Morphologically, the cancerous cell is characterized by a large nucleus, having an irregular size and shape, the nucleoli are prominent, the cytoplasm is scarce and intensely colored or, on the contrary, is pale.

As the motivation of the study is to produce virtually stained samples for further AI training or clinical diagnosis, verification on staining of cancerous samples of different kinds of cancers is vital. This should include specimens of major types of cancer, including breast cancer, lung cancer, cervical cancer and colorectal cancer, lymphoma. To demonstrate whether the proposed virtual staining approach could properly stain the tumorous tissues of the main kinds of cancer should be provided. The test set is too small for evaluation.

In evaluation, the authors should provide quantitative comparison of the methods with the reference standards (real HE stained with pathologists' annotations on the tumor regions) with respect to the Jaccard similarity coefficient. This is to demonstrate how well individual virtual staining approaches perform to stain tumour tissues / samples of various kinds of cancer.

We appreciate the reviewer's thoughtful comments and acknowledge the importance of evaluating cancerous samples. However, we wish to emphasize that the primary goal of our study is to present a general and robust method for achieving pixel super-resolution virtual staining using autofluorescence images of unlabeled tissue sections. To demonstrate this robustness and generalizability, we extensively tested our model on 180 image fields of view (FOVs) obtained from de-identified patients. **Quantitative analyses provided in Fig. 2(b), supported by statistical significance testing (t-tests), confirm that our diffusion-based virtual staining model achieves statistically significant improvements in virtual staining performance compared to the state-of-the-art.**

To further illustrate the generalizability of our proposed method, we extended our analysis to heart tissue samples, representing a distinctly different organ type. As described in the subsection titled "**Generalization to human heart tissue samples**" in our Results section, we successfully applied transfer learning using only 5 heart tissue sections and conducted blind evaluations on 178 additional FOVs from 25 new de-identified patients. **The resulting image quality metrics—PSNR, SSIM, and LPIPS—are shown in Fig. 5. The strong performance across these metrics underscores the broad applicability and robustness of our virtual staining approach across different tissue types and organs.**

In summary, on this question of the referee, we have followed the editorial recommendation: we believe that the explicit evaluation of various cancerous tissue samples and their application in clinical diagnostics is beyond the scope of this work. To clearly articulate this potential, we have now included the following paragraph in the Discussion section, quoted below:

"...Although we demonstrated the efficacy of our technique through virtual H&E staining of human lung and heart tissues, our label-free approach can be generalized to other histochemical stains and a variety of organ systems. This adaptability is reinforced by prior successes with different virtual staining methods¹. Furthermore, recent advances in virtual staining of microscopic images from cancerous samples^{9,62} underscore the potential applicability of our method to cancerous tissues, representing a promising avenue for future research. Such studies would further validate and enhance the clinical relevance and diagnostic utility of pixel super-resolution virtual staining methods."

- (2) Moreover, the authors should also perform experiments to show that with the outputs of the proposed method, how much improvements could be made using some SOTA AI models?

We sincerely thank the reviewer for this comment. To quantitatively assess the improvement our proposed method offers over existing approaches, we conducted various evaluations involving state-of-the-art (SOTA) AI models for virtual staining of label-free tissue.

First of all, we would like to emphasize that the **cGAN baseline comparisons reported in our manuscript already achieve competitive performance, establishing a strong foundation for evaluating the effectiveness of our diffusion-based virtual staining approach.** Beyond lung tissue samples, we demonstrated the success of our diffusion-based super-resolved virtual staining technique on another organ — the heart. The heart models were trained using transfer learning with data from only 5 patients and evaluated on an additional set of 178 FOVs from new de-identified patients. To showcase these results, we have the following figures and text:

- **Figure 2.** Comparison of super-resolution virtual staining performances of diffusion-based VS models and cGAN-based VS methods.
- **Figure 5.** Super-resolution virtual staining performances on human heart tissue samples using transfer learning.
- **Subsection "Generalization to human heart tissue samples"** in the Results section.

Our manuscript also includes comprehensive quantitative comparisons between super-resolved virtually stained images and their histochemically stained counterparts. In addition to SSIM and LPIPS-based image quality metric analysis, we included another quantitative metric, i.e., PSNR. Moreover, a spatial frequency-based analysis and the calculation of power spectrum cross-sections were used in our manuscript to quantitatively demonstrate the super-resolution capability of the diffusion-based VS models.

For these quantitative analyses, we have the following figures and paragraphs in our manuscript:

- **Supplementary Figure 1.** Spatial frequency spectrum analysis of virtually stained images generated by diffusion-based VS models.
- **The subsection "Super-resolved virtual staining of unlabeled tissue sections using a diffusion model"** in the Results section.
- **Figure 2.** Comparison of super-resolution virtual staining performances of diffusion-based VS models and cGAN-based VS methods.

In addition to these quantitative comparisons, we conducted a comprehensive review of the literature to establish suitable SOTA virtual staining models that can accept multiple autofluorescence (AF) channels as input for generating bright-field H&E images. The closest relevant model we identified in the literature is a recently published method (Li et al., *Nat. Commun.* DOI: 10.1038/s41467-024-46077-2, 2024) that uses two AF channels for virtual H&E staining of lung tissue. To provide a fair comparison, we evaluated the performance of this published model against our cGAN under a 1× super-resolution setting. As shown in the figure listed below, our cGAN model consistently outperformed the published model both visually and quantitatively, as measured by PSNR, SSIM, and LPIPS metrics. Furthermore, even after applying transfer learning to the published model using our training dataset, it continued to perform significantly worse than our cGAN model across all image quality evaluation metrics. **This further confirms that the cGAN baseline reported in our manuscript already achieves competitive performance, establishing a strong foundation for evaluating the effectiveness of our diffusion-based virtual staining approach.**

* Li, Y., Pillar, N., Li, J. *et al.* Virtual histological staining of unlabeled autopsy tissue. *Nat Commun* 15, 1684 (2024). <https://doi.org/10.1038/s41467-024-46077-2>

Figure caption: Comparison of virtual H&E staining performance of the published model (Y.L. *et al.*'s model) before and after transfer learning and our cGAN virtual staining model reported in the manuscript against ground truth images obtained from standard histochemical staining. (a) Visual comparisons across four individual image FOVs. (b-d) Quantitative image quality evaluations and statistical analyses using the metrics of (b) PSNR, (c) SSIM, and (d) LPIPS.

* Li, Y., Pillar, N., Li, J. *et al.* Virtual histological staining of unlabeled autopsy tissue. *Nat Commun* 15, 1684 (2024). <https://doi.org/10.1038/s41467-024-46077-2>

These findings reported above confirm that our cGAN implementation serves as a strong baseline for evaluating the effectiveness of our diffusion-based virtual staining approach. Building upon this, Figure 2 of our manuscript clearly demonstrates that our diffusion-based method not only achieves statistically significant improvements across multiple quantitative metrics (PSNR, SSIM, LPIPS), but more importantly, it also accurately reconstructs key pathological features—such as anthracotic pigment—that the cGAN baseline fails to recover. This highlights the superiority of our diffusion-based approach in virtual staining, particularly when working with spatially undersampled autofluorescence inputs.

Furthermore, following the referee’s suggestion, we also demonstrated the superiority of our diffusion-based VS framework by comparing its virtual staining performance with new SOTA models utilizing Denoising Diffusion Probabilistic Models (DDPMs)—one of the most popular architectures for image generation, editing, and enhancement. These visual and quantitative comparisons, now included in the newly added Supplementary Figure 4, further demonstrate the superior performance of our models over the DDPM-based counterparts.

We have included the following paragraph in the Discussion section of our revised manuscript to highlight the results of these new comparisons:

*“...Denoising Diffusion Probabilistic Models^{20,21} (DDPM) are among the most widely used and powerful diffusion frameworks for image generation^{20,21,24}, editing^{23,61}, and enhancement⁴⁹. Their recent success in stain transformation related tasks^{29,30} highlights their potential in computational pathology. To further demonstrate the effectiveness of our diffusion-based VS models (D_x), we compared their performance against DDPM-based VS models (P_x), as shown in **Supplementary Figure 4**. To provide a fair comparison, the DDPM-based models employed the original DDPM sampling process²⁰ and used the same denoising U-Net architecture as our models. **Quantitative evaluations on blind test data, presented in Supplementary Figures 4(b–c), clearly show that our models outperform the DDPM-based virtual staining counterparts, underscoring the effectiveness of our proposed approach.**”*

We have also included below the newly added Supplementary Figure 4:

Supplementary Figure 4. Comparative evaluation of pixel super-resolution virtual staining performance of our diffusion VS models against DDPM-based models. (a) Visual comparisons of virtually stained H&E images generated by our diffusion VS models (top row) and the DDPM-based diffusion VS models (bottom row). Each model was independently trained and evaluated for specific super-resolution factors ranging from 1× to 5×. Evaluations were conducted using autofluorescence images from 180 distinct FOVs obtained from 15 unlabeled lung samples. (b) Bar plots illustrating quantitative comparisons of SSIM, LPIPS, and PSNR metrics, averaged across all test images virtually stained by our diffusion VS models and the DDPM-based VS models. Error bars indicate the standard error of the mean. Labels D_x and P_x represent our VS models and DDPM-based VS models, respectively, for each super-resolution factor x . (c) Bar plots depicting t -scores comparing the performance differences between our VS models and the DDPM-based VS models at identical super-resolution factors. Green shaded areas highlight statistically significant improvements in virtual staining performance achieved by our VS models relative to the DDPM-based VS models.

- (3) My suggestion is similar to the second reviewer that there's no quantification evaluation for pathologists reviewing the virtually stained outputs to confirm they're biologically meaningful. More importantly, the major issue of this study is the lack of cancerous samples in the experiments. To demonstrate whether the proposed virtual staining approach could properly stain the tumorous tissues of the main kinds of cancer should be provided. The test set is too small for evaluation.

As mentioned in our response to Comment #1, **on the tumorous tissue-related question of the referee, we have followed the editorial recommendation: we believe that the explicit evaluation of various cancerous tissue samples and their application in clinical diagnostics is beyond the scope of this work.** To clearly articulate this potential, we have now included the following paragraph in the Discussion section, quoted below:

"...Although we demonstrated the efficacy of our technique through virtual H&E staining of human lung and heart tissues, our label-free approach can be generalized to other histochemical stains and a variety of organ systems. This adaptability is reinforced by prior successes with different virtual staining methods¹. Furthermore, recent advances in virtual staining of microscopic images from cancerous samples^{9,62} underscore the potential applicability of our method to cancerous tissues, representing a promising avenue for future research. Such studies would further validate and enhance the clinical relevance and diagnostic utility of pixel super-resolution virtual staining methods."

Furthermore, a board-certified pathologist (N.P.) assessed the virtual staining results presented in Figures 2–5 and provided several additional comparisons between the diffusion-model generated images and their corresponding histochemically stained counterparts, confirming that the virtually stained outputs faithfully capture key biological and pathological structures. Accordingly, we have added the following statements to the main text to highlight these findings:

"...This aligns with the comparison of the diffusion model-generated images and their matched histochemical counterparts, conducted by a board-certified pathologist (N.P.), which demonstrated complete concordance across all image subsegments (fibro-collagenous stroma, blood vessels, immune cells, and anthracotic pigment)."

"...The virtually stained images appear highly similar to the histochemically stained image, effectively representing lung morphology, including small airways and capillaries."

"...The virtually stained heart images provided an accurate representation of cardiac myocytes and the interstitium, clearly visualizing muscle striations and intercalated discs in longitudinal sections, as well as centrally located nuclei in cross sections."

Additionally, we revised the caption of Figure 3 to include:

"...An assessment conducted by a certified pathologist (N.P.) revealed strong structural similarity across all image subsegments (e.g., alveoli, blood vessels, and scattered lymphocytes)."

Remarks on code:

Please provide clear instructions on the Github web page for all training, testing and evaluation process. So far, training and evaluation codes are missing, which will be hard for reproducibility.

We have uploaded the training and evaluation codes to GitHub for public release.

To conclude, we sincerely thank the referees for their constructive comments and feedback, which helped us to further improve the quality and clarity of our manuscript. We look forward to hearing back from you regarding our revised submission.